# HyPER: Bridging Exploration and Exploitation for Scalable LLM Reasoning with Hypothesis Path Expansion and Reduction

**Shengxuan Qiu** [* 1 2]  **Haochen Huang** [* 1 2]  **Shuzhang Zhong** [1 2]  **Pengfei Zuo** [3]  **Meng Li** [1 2]

## Abstract

Scaling test-time compute with multi-path chain-of-thought can improve reasoning accuracy, but its gains hinge on an effective exploration–exploitation trade-off. Existing methods handle this trade-off in rigid ways: tree-structured search hard-codes exploration via brittle expansion rules that disrupt post-trained reasoning, while parallel reasoning over-explores redundant hypothesis paths and relies on a weak answer selection strategy. Driven by the insight that the optimal balance is *phase-dependent* and that correct vs. incorrect paths often *diverge only at late stages*, we reconceptualize test-time scaling as a dynamic *expand–reduce* control problem over a pool of hypothesis paths. We introduce *HyPER*, a *training-free online control policy* for MoE multi-path decoding that reallocates compute under a fixed budget using lightweight path statistics. HyPER features (i) an *online controller* that shifts from exploration to exploitation as the hypothesis pool evolves, (ii) an MoE-based token-level refinement primitive for efficient *generation-time exploitation* without full-path resampling, and (iii) a length- and confidence-aware aggregation rule to bridge the existence–selection gap for reliable *answer-time exploitation*. Extensive experimental results across four MoE models and diverse benchmarks demonstrate HyPER consistently achieves the accuracy–compute Pareto frontier, outperforming prior-art methods by $8 \sim 10\%$ while reducing token consumption by $25 \sim 40\%$.

## 1. Introduction

Large language models (LLMs) have demonstrated remarkable reasoning capabilities when prompted to produce chain-of-thought (CoT) (Wei et al., 2022). However, a single CoT remains brittle on complex problems (Wang et al., 2022; Zhao et al., 2026). Scaling *test-time compute* mitigates this fragility by exploring multiple hypothesis reasoning paths simultaneously to maximize the probability of success (Zhang et al., 2025). Fundamentally, this process necessitates a delicate balance between *exploration* (i.e., generating diverse hypotheses) and *exploitation* (i.e., refining promising hypotheses) under strict compute budgets (Snell et al., 2024; Ding et al., 2025).

Existing test-time scaling methods typically fall into two paradigms, yet both struggle to maintain this balance. Test-time scaling can be instantiated at three *granularities*: token (a single decoding iteration), step (a short reasoning chunk spanning multiple tokens), and path (a complete CoT path from prompt to final answer). Tree-based search schemes explicitly grow intermediate steps and use step-level rewards to guide branching (i.e., expanding multiple candidate next steps) (Yao et al., 2023; Xie et al., 2024; Snell et al., 2024; Hooper et al., 2025). However, they enforce a rigid exploration schedule that requires branching after predefined reasoning steps. This not only wastes compute on unnecessary exploration, but also disrupts the native continuous reasoning flow of post-trained models (Lian et al., 2025; Yang et al., 2025).

In contrast, *parallel reasoning*, e.g., Self-Consistency and Best-of-$N$, samples multiple complete CoT paths and aggregates their results (Wang et al., 2022; Brown et al., 2024; Irvine et al., 2023). It preserves native generation, but the balance tilts *heavily toward exploration*: much of the budget is spent on redundant near-duplicate paths, and it offers limited *test-time exploitation* which aims to improve the reasoning quality. Recent *adaptive* parallel reasoning variants aim to improve this trade-off, yet typically adjust only part of the equation. For example, *spawn–join* methods introduce more structured exploration by dynamically branching and joining sub-reasoning steps, but their exploitation behavior is usually *learned* and thus requires specialized training (e.g., SFT or RL) to decide when to branch and how to

---

[*]Equal contribution  [1]Institute for Artificial Intelligence, Peking University, Beijing [2]School of Integrated Circuits, Peking University, Beijing [3]Huawei. Correspondence to: Meng Li <meng.li@pku.edu.cn>, Pengfei Zuo <pfzuo.cs@gmail.com>.

*Proceedings of the $43^{rd}$ International Conference on Machine Learning*, Seoul, South Korea. PMLR 306, 2026. Copyright 2026 by the author(s).

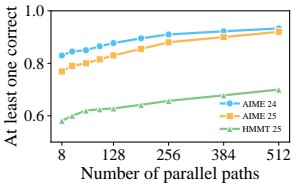
*(a)* Existence Probability.

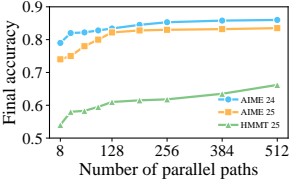
*(b)* Final Accuracy.

*Figure 1.* Both the existence probability and the accuracy show an increasing trend as the number of paths increases, yet there is a significant marginal benefit.

join (Lian et al., 2025; Wang et al., 2025a; Yu et al., 2025). Training-free approaches such as DeepConf (Fu et al., 2025) make exploration cheaper via pruning, but remain *purely pruning*: they mainly *reduce exploration* (filter low-quality paths) without explicitly reallocating the saved budget into stronger *exploitation*, leaving promising hypotheses under-refined and still relying on large initial widths for coverage.

Recently, reasoning models increasingly adopt *Mixture-of-Experts* (MoE) architectures, which offer inherent routing diversity for token-level refinement (e.g., (Zibakhsh et al., 2025)). While this unlocks efficient local improvement, token-level gains alone do not resolve the global exploration–exploitation trade-off: deciding *when* to widen the search or refine a hypothesis still requires dynamic allocation.

Accordingly, we shift the focus from static search schedules to *closed-loop* resource control. Our design is motivated by three key empirical observations about the *dynamic* nature of this balance:

1. **Phase-dependent utility of exploration.** As shown in Figure 1, increasing path width improves coverage but yields diminishing returns on final accuracy. Crucially, this utility is *phase-dependent*: early exploration secures coverage, whereas late-stage redundancy merely consumes budget. This indicates that a static schedule is inefficient, creating an opportunity to improve performance by *dynamically reallocating* compute from exploration to exploitation based on the decoding state.

2. **Correct and incorrect paths often diverge late.** Figure 2 provides dataset-level evidence on AIME25 and HMMT25. For each generated path, we normalize its reasoning trajectory by length, compute token confidence at each relative position, and average the resulting curves over correct and incorrect paths across questions. The trends show that correct and incorrect paths have similar confidence patterns in early decoding but separate near the tail: correct paths remain relatively stable, whereas incorrect paths degrade. This suggests that full-path resampling is often wasteful for correcting late-stage errors; instead, compute can be used more efficiently through *local* refinement around low-confidence tail regions.

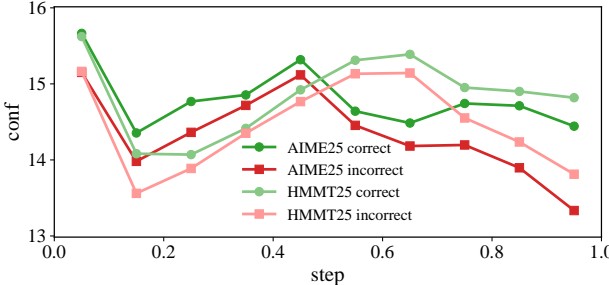

*Figure 2.* Dataset-level confidence divergence between correct and incorrect reasoning paths. Each path is normalized by length; token confidence is averaged at each relative position over correct and incorrect paths on AIME25 and HMMT25.

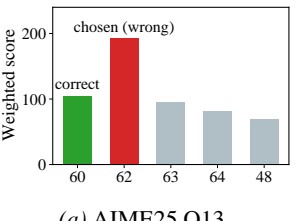
*(a)* AIME25 Q13.

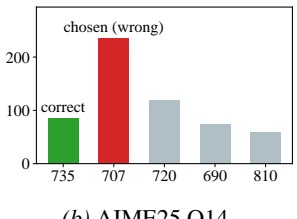
*(b)* AIME25 Q14.

*Figure 3.* Correct paths are present but outvoted by noisy paths under confidence-weighted voting: each path's answer receives a weight given by the path's global average token confidence.

3. **The existence–selection gap.** Even when multi-path decoding succeeds in generating a correct candidate, answer selection remains fragile. As demonstrated in Figure 3, frequent but incorrect answers can dominate naive majority voting—and even confidence-weighted voting—leading to mis-selection even when a correct solution is present. This highlights the necessity for *answer-time* exploitation mechanisms that incorporate structural reliability signals beyond simple frequency to bridge the gap between existence and selection.

We therefore introduce *HyPER* (**Hy**pothesis **P**ath **E**xpansion and **R**eduction), a *training-free online control policy* that reconceptualizes test-time scaling as a dynamic *expand–reduce control problem* over a pool of hypothesis paths. Instead of following a rigid schedule, HyPER employs a closed-loop controller that monitors lightweight real-time path statistics to reallocate compute budget on the fly.

HyPER makes three core contributions: (i) An *online expand–reduce controller* that coordinates exploration and exploitation by dynamically selecting among path branching (*Branch*), short-horizon exploration (*MultiToken*), and token-level refinement (*SingleToken*) based on the pool's diversity and confidence state. (ii) An efficient *single-token aggregation primitive* that leverages *MoE* routing diversity to produce and aggregate complementary token-level hypotheses, enabling strong generation-time exploitation without full-path resampling. (iii) A *length- and confidence-aware*

*voting rule* that addresses the existence–selection gap by exploiting the pruning-induced length bias for reliable answer selection. HyPER requires no fine-tuning and integrates seamlessly with post-trained reasoning models.

Empirically, across four MoE LLMs and a diverse set of reasoning benchmarks, HyPER improves accuracy by an average of 8–10 percentage points over existing test-time scaling baselines, while reducing token usage by approximately 25–40% under comparable compute budgets.

## 2. Background

**Paradigms of test-time scaling.** LLM reasoning increasingly benefits from additional *test-time compute* (Snell et al., 2024; Welleck et al., 2024). Across both "depth" (longer internal deliberation within a single CoT) (OpenAI et al., 2024; Guo et al., 2025; Kimi et al., 2025; xAI, 2025; Muennighoff et al., 2025; Ye et al., 2025) and "breadth" (searching over multiple hypotheses), test-time scaling can be viewed as *inference-time search* that must trade off exploration and exploitation under finite budgets. Concretely, methods differ by (i) whether they perform *adaptive control* over exploration/exploitation online or follow a rigid schedule, (ii) what *exploitation primitives* they employ: pure final *answer evaluation* (e.g., voting/selection), confidence-based *pruning* to reduce wasted exploration, or finer-grained *token-level refinement*, and (iii) whether guidance comes from the *policy* model alone (the base generator used to expand paths) or additionally from a learned *reward* model that assigns step- or answer-level scores.

Existing approaches mainly fall into two paradigms. *Structured tree-based search* methods explicitly expand intermediate reasoning states and rely on step-level scoring or external supervision (e.g., ToT, MCTS, ETS) (Yao et al., 2023; Xie et al., 2024; Snell et al., 2024; Hooper et al., 2025). Such methods instantiate exploration via branching and exploitation via score-guided selection, but typically depend on an auxiliary verifier (and thus require additional training and can be rigid and brittle for post-trained reasoning models (Appendix H). In contrast, *parallel reasoning* methods sample multiple complete trajectories and aggregate only at the end, as in Self-Consistency and Best-of-$N$ (Wang et al., 2022; Brown et al., 2024; Irvine et al., 2023). They preserve native generation behavior and are often policy-only (no verifier), but tend to over-explore redundant paths and under-utilize promising hypotheses during generation. Recent *adaptive variants* allocate test-time compute online: some focus on spawn-join execution (Lian et al., 2025), while others, such as DeepConf (Fu et al., 2025), introduce training-free confidence-guided pruning to adapt the active pool under a fixed budget. However, many such variants are primarily subtractive (prune-heavy): they lack the ability to exploit dedicated hypothesis and thus, rely on extending the

*Table 1.* TTS methods as checklists. Adap ctrl: online expand/refine decisions. Answer eval: aggregate and select the answer over multi-paths.

| Method | Gran. | Exploration | | Exploitation | | | Train-free | |
|---|---|---|---|---|---|---|---|---|
| | | Adap ctrl | No rigid search | Answer eval | Prune | Token refine | Policy model | Reward model |
| *Tree-based search schemes* | | | | | | | | |
| ToT (Yao et al., 2023) | step | ✗ | ✗ | ✓ | ✗ | ✗ | ✓ | ✗ |
| MCTS (Xie et al., 2024) | step | ✓ | ✗ | ✓ | ✗ | ✗ | ✓ | ✗ |
| ETS (Hooper et al., 2025) | step | ✓ | ✗ | ✓ | ✓ | ✗ | ✗ | ✗ |
| *Parallel reasoning* | | | | | | | | |
| SC (Wang et al., 2022) | path | ✗ | ✓ | ✓ | ✗ | ✗ | ✓ | ✗ |
| BoN (Brown et al., 2024) | path | ✗ | ✓ | ✓ | ✗ | ✗ | ✓ | ✗ |
| *Adaptive reasoning* | | | | | | | | |
| DeepConf (Fu et al., 2025) | path | ✓ | ✓ | ✓ | ✓ | ✗ | ✓ | ✓ |
| Thread (Lian et al., 2025) | step | ✓ | ✓ | ✓ | ✗ | ✗ | ✗ | ✓ |
| *No multi-paths* | | | | | | | | |
| RoE (Zibakhsh et al., 2025) | token | ✗ | ✓ | ✗ | ✗ | ✓ | ✓ | ✓ |
| *Expand+reduce* | | | | | | | | |
| **HyPER (ours)** | token | ✓ | ✓ | ✓ | ✓ | ✓ | ✓ | ✓ |

number of initial hypotheses to improve task accuracy.

**Token-level test-time scaling in MoE.** While standard scaling methods operate at the *path level*, Mixture-of-experts (MoE) architectures provide a latent dimension for *token-level* scaling. By routing tokens to sparse expert subsets, MoEs inherently contain a diverse ensemble of submodels. Recent work, such as RoE (Zibakhsh et al., 2025), exploits this by injecting Gumbel noise into expert routing to generate multiple expert proposals per token and aggregating them into a single prediction, effectively turning an MoE into a dynamic ensemble at inference time. However, existing token-level methods operate in isolation: they are typically "always-on" and agnostic to the broader reasoning state. They operate in isolation, blind to the trade-off between refining the current token and exploring new paths. This leaves an open challenge: *how to dynamically allocate a unified budget between token-level refinement and path-level expansion.*

For context, Table 1 summarizes representative test-time scaling methods by decision granularity and how they instantiate exploration and exploitation.

## 3. HyPER Design

Figure 4 provides an overview of our HyPER framework. HyPER instantiates test-time scaling as *online expand–reduce control* over a pool of hypothesis paths, and couples three components to balance exploration and exploitation under a fixed budget: (i) a lightweight controller that periodically selects among branching, multi-token aggregation, and single-token aggregation actions during decoding (§3.1); (ii) a token-level refinement primitive (*single-token aggregation*) that exploits MoE routing diversity without resampling

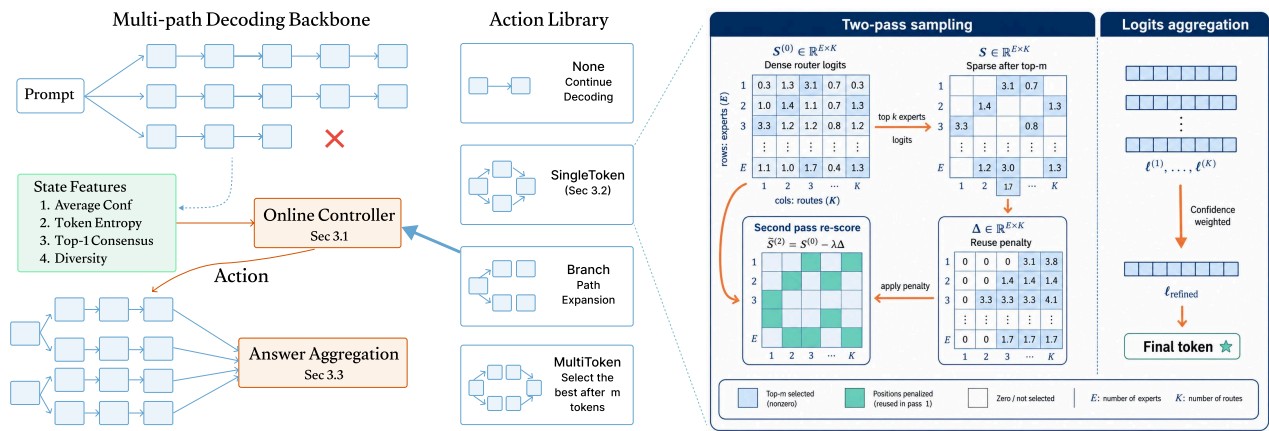

*Figure 4.* Overview of HyPER.

full paths (§3.2); and (iii) a length-aware, confidence-guided voting rule that strengthens answer-time exploitation (§3.3). Throughout decoding, an always-on confidence pruning filter continuously removes low-quality paths, while the controller acts only on the remaining survivors.

**Online signals for confidence and diversity.** Effective online control relies on lightweight, training-free signals to estimate the quality and diversity of the surviving path pool $\mathcal{S}_t$. We choose signals that are (i) inexpensive to compute online, (ii) monotonic with respect to path reliability or diversity, and (iii) already used or validated in prior test-time scaling work.

*Confidence signals.* We track mean token confidence $\bar{C}_t$, mean token entropy $H_t$ as an uncertainty proxy (via a top-$k$ approximation), and top-1 consensus $\beta_t$, defined as the vote share of the most common next token across paths, following DeepConf (Fu et al., 2025).

*Diversity signals.* To detect path collapse and trigger exploration, we combine two complementary similarity measures commonly used for LLM outputs: distribution-level divergence $D_{\mathrm{dist}}$ and suffix-level edit distance $D_{\mathrm{seq}}$ (Xia et al., 2025; Zheng et al., 2025). They are aggregated into a single diversity score $D_t = \eta\, D_{\mathrm{dist},t} + (1-\eta)\, D_{\mathrm{seq},t}$, where low $D_t$ indicates collapsing trajectories and favors branching or multi-token aggregation, while higher $D_t$ suggests sufficient coverage to prioritize refinement. Complete metric definitions and computation details are deferred to Appendix A.

## 3.1. Online Path Expansion and Reduction

To operationalize the phase-dependent exploration–exploitation trade-off (Observation 1), we rely on the insight that decoding exposes evolving information through *lightweight online signals*. Inexpensive metrics like confidence and diversity serve as effective, training-free proxies for the pool's state—signaling when to trigger

exploration (e.g., upon diversity collapse) or exploitation (e.g., under high uncertainty). Accordingly, we maintain a pool of *surviving* paths $\mathcal{S}_t$ subject to always-on confidence pruning. Every $T$ decoding steps, the controller evaluates these statistics and outputs an action $a_t \in \{\textsc{None}, \textsc{SingleToken}, \textsc{MultiToken}, \textsc{Branch}\}$ to apply to all survivors. The complete HyPER decoding pipeline is detailed in Algorithm 1 (Appendix B).

### 3.1.1. RUNTIME SIGNALS FOR ADAPTIVE CONTROL

At each decision point $t$, we compute confidence-group signals $\{\bar{C}_t, H_t, \beta_t\}$ (mean confidence, mean top-$k$ entropy with $k = 8$, and top-1 consensus; following DeepConf) and a diversity signal $D_t$ that measures whether paths are collapsing (Section 2). We map all statistics to $[0, 1]$ via global-max normalization estimated during warm-up: in formulas below we use $\min(X_t/X_{\max}, 1)$, where $X_{\max}$ is the maximum observed value of $X_t$ in warm-up (Section 4.1). Although the diversity signals involve pairwise comparisons with $O(|S_t|^2)$ complexity, the survivor pool is capped by $S_{\max} = 80$ and the controller acts only every $T$ steps, making the amortized overhead negligible in practice.

### 3.1.2. ACTION SET

We consider four actions $a_t \in \{\textsc{None}, \textsc{SingleToken}, \textsc{MultiToken}, \textsc{Branch}\}$. At decision step $t$, let $S_t = |\mathcal{S}_t|$ be the current number of surviving paths. To maintain effective parallelism as pruning shrinks the pool, we set the per-path expansion factor to $r_t = \lceil W/S_t \rceil$, where $W$ is the target (initial/budgeted) pool width. Thus, each surviving path expands to roughly $r_t$ parallel continuations (subject to a global width cap when needed).

**NONE (continue standard decoding).** NONE performs no additional operation: we continue decoding all active paths. Intuitively, this action is preferred when the pool is al-

ready reliable (high $\bar{C}_t$), confident (low $H_t$), and maintains sufficient coverage (high $D_t$), making additional intervention unnecessary.

**SINGLETOKEN (single-token expand-and-aggregate; exploitation).** SINGLETOKEN applies a token-level expand–reduce operation (Sec. 3.2) to each active path, refining the next-token decision without changing the number of paths. This action targets *test-time* exploitation and activates under local instability (low confidence, high entropy, low consensus), where token-level refinement is most effective (see Figure 5).

**BRANCH (pure branching; exploration).** BRANCH explicitly widens the pool: each surviving path forks into $r_t$ child paths that continue decoding independently. This action corresponds to pure exploration and triggers when diversity is insufficient (low $D_t$) and paths disagree (low consensus), increasing coverage of the hypothesis space (Figure 5).

**MULTITOKEN ($m$-token expand-and-aggregate; hybrid).** MULTITOKEN applies short-horizon expansion followed by aggregation (we fix $m{=}T$). Starting from each surviving root path, we spawn $r_t$ child continuations and decode them for the next $m$ tokens. We then *aggregate back per root*: among the $r_t$ children sharing the same root, we keep only the child with the highest window-level confidence over these $m$ steps, discard the others, and release their KV caches. As reflected in Figure 5, this action plays a hybrid role: it activates when diversity is low but the pool exhibits strong confidence dispersion (high $\mathrm{Var}(C)_t$), making short-horizon expand–reduce effective.

3.1.3. ACTION SELECTION

Note that the coefficients in these equations are simple averaging weights, reflecting our design philosophy that coarse, lightweight signals are sufficient for effective control without learning complex policies. At decision step $t$, the controller computes one score per action and selects the highest-scoring action. We briefly write the normalized statistics as $\widehat{X}_t = \min(X_t/X_{\max}, 1)$.

$$\text{Score}_{\text{NONE}}(t) = \frac{1}{3}\left(\widehat{\bar{C}}_t + (1 - \widehat{H}_t) + \widehat{D}_t\right), \quad (1)$$

$$\text{Score}_{\text{SINGLETOKEN}}(t) = \frac{1}{3}\left((1 - \widehat{\bar{C}}_t) + \widehat{H}_t + (1 - \beta_t)\right), \quad (2)$$

$$\text{Score}_{\text{BRANCH}}(t) = \frac{1}{2}\left((1 - \widehat{D}_t) + (1 - \beta_t)\right), \quad (3)$$

$$\text{Score}_{\text{MULTITOKEN}}(t) = \frac{1}{3}\left((1 - \widehat{D}_t) + (1 - \widehat{\bar{C}}_t) + \widehat{\mathrm{Var}(C)}_t\right), \quad (4)$$

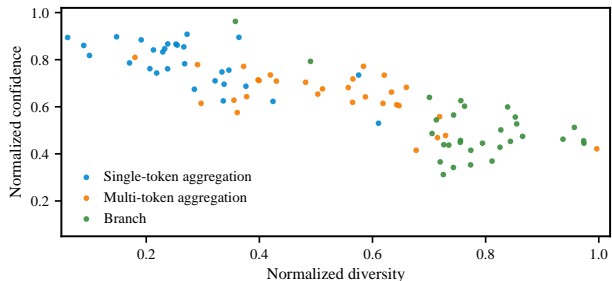

*Figure 5.* Per-instance confidence–diversity scatter plots under isolated actions.

The coefficients (e.g., $1/3$) are simply normalization factors $1/n$ for the $n$ signals per action, not tuned hyperparameters. This minimalist averaging ensures all signals contribute equally without dataset-specific tuning. As validated in Figure 8, this simple logic successfully captures the phase-dependent shift from exploration to exploitation.

### 3.2. Single-Token Expand-and-Aggregate

Addressing the finding that correct and incorrect paths often diverge late and are best separated by tail confidence (Observation 2), this module targets efficient *test-time exploitation* without the overhead of resampling full paths. We instantiate SINGLETOKEN as a token-level *expand–reduce* operator that leverages the latent routing diversity of MoE models. For the current token, we *expand* the computation into $K = r_t = \lceil W/S_t \rceil$ stochastic expert-routed proposals and then *reduce* them into a single refined logit vector via a reuse-aware two-pass sampler and confidence-weighted aggregation (Figure 4). Crucially, this mechanism stabilizes the tail confidence needed for correctness, while enforcing *intra-token diversity* to prevent refinement from collapsing path-level exploration.

**Step 1: Gumbel-noise route sampling.** To initiate the expansion, we first perturb the MoE router logits $\mathbf{r} \in \mathbb{R}^E$ for the current token. We draw $K$ Gumbel noise vectors $\mathbf{g}^{(k)} \in \mathbb{R}^E$ with $g_e^{(k)} \sim \mathrm{Gumbel}(0,1)$, and construct a dense score matrix $S^{(0)} \in \mathbb{R}^{E \times K}$ where:

$$S_{:,k}^{(0)} = \mathbf{r} + \tau\,\mathbf{g}^{(k)}, \qquad k \in \{0, \ldots, K-1\}, \quad (5)$$

with $\tau = 0.5$ controlling the exploration strength.

**Step 2: Two-pass expert sampling.** To enforce intra-token diversity, we feed $S^{(0)}$ into a fully vectorized two-pass sampling scheme rather than selecting experts independently. In the first pass, we apply standard top-$m$ routing (where $m$ is the backbone's fixed gate size) to each column of $S^{(0)}$, yielding a sparse matrix $S$. To discourage expert

collisions across the $K$ routes, we build a penalty matrix $\Delta \in \mathbb{R}^{E \times K}$ via a *column-wise* prefix accumulation. Setting the first column to zero ($\Delta_{:,0} = \mathbf{0}$), subsequent columns aggregate prior expert usage via $\Delta_{:,k} = \phi\left(\sum_{t=0}^{k-1} S_{:,t}\right)$ using a smooth squashing function $\phi(\cdot)$ (e.g., $\tanh$). In the second pass, we reuse the original dense scores and apply this deterministic penalty: $\widetilde{S}^{(2)} = S^{(0)} - \lambda \Delta$. A final top-$m$ routing on $\widetilde{S}^{(2)}$ yields the diversified expert selections. Pseudo-code are provided in Appendix C.

**Step 3: Confidence-weighted logit aggregation.** Once the $K$ diverse expert routes are forwarded, we obtain $K$ candidate next-token logit vectors. To reduce them into a single reliable prediction, we apply confidence-weighted aggregation. Each route inherits a *token confidence score $C_t$* derived from its predictive distribution (Equation 6), where higher $C_t$ indicates lower uncertainty. We convert these scores into smoothed weights to softly combine the candidate logits, amplifying confident routes while reducing variance and preventing the diversity penalty from degrading generation quality. While involving multiple passes, these vectorized matrix operations impose negligible latency compared to generating a new token. By forcing local diversity without full-path resampling, SINGLETOKEN offers a high-efficiency exploitation primitive.

**Memory optimization via KV cache sharing.** Computationally, naive single-token expansion would multiply memory costs by $K$. To avoid this, we share the prefix KV states across the $K$ routed proposals and localize stochastic routing only to the current token. Thus, KV memory scales strictly with the number of surviving paths (capped by $S_{\max}$) rather than with $K$ (Zibakhsh et al., 2025). Details are provided in Appendix D.

### 3.3. Answer Aggregation with Length- and Confidence-Aware Voting

To address the *Existence–Selection Gap* (Observation 3), where having at least one correct path does not guarantee the correct final answer. We identify a crucial structural signal for *answer-time exploitation*. As shown in Figure 6 and further analyzed in Appendix E.2, **confidence pruning inverts the typical length bias**: *surviving correct paths become systematically longer* than incorrect ones, as erroneous reasoning is frequently exposed by low-confidence steps and terminated early.

Leveraging this asymmetry, we collect all surviving and pruned paths that produced a scalar answer. Let $\mathcal{P}$ denote this set; for each path $p \in \mathcal{P}$, let $a_p$ be its answer, $L_p$ its length, and $\hat{c}_p$ its global average confidence. We emphasize that length serves as a robust signal *only* when coupled with confidence pruning, rather than as a universally valid metric (verified in Appendix E.3). We first re-

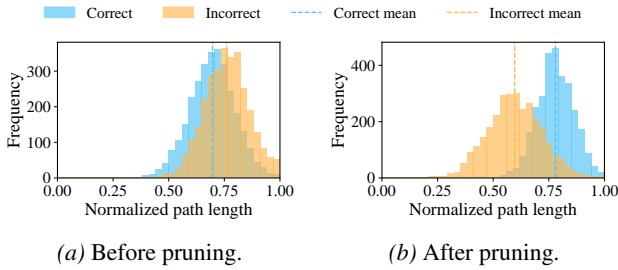

*(a) Before pruning.*    *(b) After pruning.*

*Figure 6.* Length asymmetry induced by confidence pruning.

strict attention to the top-$K_a$ answers by majority count, denoted as $\mathcal{A}_K = \text{TopK}_a\left(\sum_{p \in \mathcal{P}} \mathbf{1}[a_p = a]\right)$. For each candidate answer $a \in \mathcal{A}_K$, we assign a support score $s(a) = \sum_{p \in \mathcal{P}: a_p = a} (\lambda_{\text{len}} L_p / Z_L + \lambda_{\text{conf}} \hat{c}_p / Z_c)$, and select $\hat{a} = \arg\max_{a \in \mathcal{A}_K} s(a)$, where $Z_L = \sum_{q \in \mathcal{P}} L_q$ and $Z_c = \max_{q \in \mathcal{P}} \hat{c}_q$. Here $\lambda_{\text{len}}$ and $\lambda_{\text{conf}}$ control the relative contributions of rationale length and confidence. This rule is compatible with confidence pruning and expand–reduce control, and admits a Bayes-optimal interpretation under a simple generative model (Appendix F). We also empirically verify robustness against *long but incorrect* paths in Appendix E.4.

## 4. Experiments

### 4.1. Experimental Setup

**Decoding, budget, and accounting.** All methods use the same prompts, answer extraction, SC-style multi-path decoding backend, and sampling hyperparameters, with an initial/concurrent path cap of $S_{\max} = 80$ unless stated otherwise. We apply a DeepConf-style warm-up pruning scheme: draw $N_{\text{init}} = 16$ initial SC traces, estimate a sliding-window confidence threshold from the top-10 traces, and terminate any path whose confidence falls below the threshold thereafter (Fu et al., 2025).

*Budget alignment.* Comparing methods at the same initial width would undercharge HyPER, because BRANCH and MULTITOKEN instantiate transient continuations beyond the surviving path pool. We therefore align methods by their *Total Instantiated Path Count* ($N_{\text{inst}}$), which counts all paths instantiated during decoding, including transient branches. Since HyPER typically yields $N_{\text{inst}} \approx 1.5 S_{\max}$, we run SC and DeepConf with an equivalently enlarged initial width ($1.5 S_{\max}$) to match the effective path budget. We report token cost normalized by the standard SC-80 baseline ($1.0\times$). For RoE and SINGLETOKEN, each expert-routed proposal is charged as $K$ effective token expansions. In the Pareto analysis (Figure 7), we sweep $S_{\max} \in \{32, 64, 80, 128, 256, 512\}$ while maintaining the same effective-budget accounting. Thus, improvements cannot be attributed to simply using more sampled paths or uncharged expert proposals.

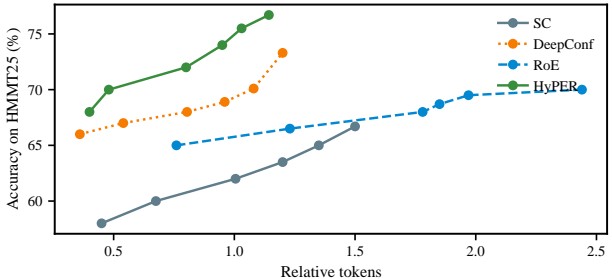

*Figure 7.* Accuracy–compute Pareto curves on HMMT25 under varying budgets $S_{\max} \in \{32, 64, 80, 128, 256, 512\}$. The relative token cost is normalized by the SC baseline at budget $S_{\max} = 80$.

Unless stated otherwise, we use $\eta = 0.4$ for the diversity mixture, controller interval $T = 64$, SINGLETOKEN reuse penalty $\lambda = 0.1$, and answer aggregation weights $(\lambda_{\text{len}}, \lambda_{\text{conf}}) = (0.6, 0.4)$. Appendix E.1 provides sensitivity analyses for these hyperparameters.

**Benchmarks and models.** We evaluate HyPER on two tiers of MoE reasoning settings. For hard reasoning, we test **Qwen3-30B-A3B-Thinking-2507** and **Qwen3-Next-80B-A3B-Thinking** (Yang et al., 2025) on AIME24/25 (Mathematical Association of America, 2024; 2025), HMMT25 (HMMT Organization, 2025), and HLE (Phan et al., 2025), using official scoring. For light reasoning, we test **OLMoE-1B-7B-0924-Instruct** (Muennighoff et al., 2024) and **DeepSeek-V2-Lite-Chat** (DeepSeek-AI, 2024) on Math500 (Hendrycks et al., 2021), GSM8K (Cobbe et al., 2021), ARC-C, and ARC-E (Clark et al., 2018). Baselines include SC, Self-Certainty (Kang et al., 2025), DeepConf (Fu et al., 2025), and late-stage RoE (Zibakhsh et al., 2025), all under the same prompting and decoding setup.

## 4.2. Main Results

Table 2 reports accuracy and normalized token cost across all evaluated models and datasets under the effective-budget-aligned setup. Overall, HyPER consistently improves the accuracy–compute trade-off over SC, Self-Certainty, Deep-Conf, and RoE. The gains are most pronounced on hard multi-step benchmarks, where pruning-only methods reduce wasted exploration but do not explicitly refine promising paths, while always-on token-level refinement improves local token quality at substantially higher effective token cost. By contrast, HyPER adaptively reallocates budget across pruning, branching, short-horizon expansion, token-level refinement, and answer aggregation, leading to higher accuracy with lower or comparable normalized token cost.

To further characterize the trade-off across different inference budgets, Figure 7 plots Pareto curves on HMMT25 by sweeping $S_{\max} \in \{32, 64, 80, 128, 256, 512\}$, with token cost normalized by the SC baseline at budget $S_{\max} = 80$. Across

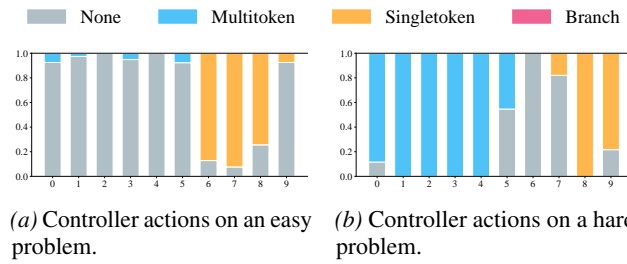

*(a)* Controller actions on an easy problem.

*(b)* Controller actions on a hard problem.

*Figure 8.* Controller behavior in different cases.

budgets, HyPER remains on or near the Pareto frontier, showing that its gains are not tied to a single operating point. Instead, the controller consistently converts additional test-time compute into useful exploration or exploitation.

**Architecture generality.** Although the main results focus on MoE backbones because SINGLETOKEN exploits expert-routing diversity, the online expand–reduce controller itself is architecture-agnostic. Appendix G provides a dense-model controller-only demonstration, showing that the adaptive control component can improve reasoning even without MoE-specific token refinement.

## 4.3. Controller Behavior and Adaptive Exploration

We next examine how the HyPER controller allocates actions over time. Figure 8 plots the action trajectories for two representative questions: an "easy" problem that most paths solve quickly, and a "hard" problem on which self-consistency struggles. For the easy question (Fig. 8a), the controller quickly settles on NONE and only occasionally invokes SINGLETOKEN, rarely triggering BRANCH or MULTITOKEN, matching our goal of avoiding unnecessary exploration when signals are already confident and consistent. For the hard question (Fig. 8b), the controller behaves differently: it activates BRANCH and MULTITOKEN more often to widen the hypothesis set, and then gradually shifts to SINGLETOKEN to exploit high-confidence paths later.

Beyond these case studies, we also report dataset-level aggregate action frequencies for Qwen3-30B on two representative benchmarks (AIME24/HLE) in Figure 9. As tasks become harder, the fraction of NONE decreases while the controller more frequently triggers exploration (BRANCH/MULTITOKEN) and exploitation (SINGLETOKEN), indicating that HyPER adapts its compute allocation to problem difficulty.

## 4.4. Two-Pass Sampling versus RoE

Figure 10 compares our SingleToken two-pass refinement against a RoE-style uniform token-level refinement baseline. Panel (a) shows that both methods substantially improve tail-token confidence over no refinement, indicating that token-level logit aggregation is an effective exploitation

*Table 2.* Token cost and accuracy across models and datasets. **Token** is normalized by **SC** within each (model, dataset).

| Model | Dataset | SC | | Self-Certainty | | DeepConf | | RoE | | HyPER | |
|---|---|---|---|---|---|---|---|---|---|---|---|
| | | Token | Acc | Token | Acc | Token | Acc | Token | Acc | Token | Acc |
| **Qwen3-30B** | AIME24 | 1.00 | 88.0 ($\pm$ 2.0) | 0.94 | 83.3 ($\pm$ 3.3) | 0.46 | 92.7 ($\pm$ 2.7) | 1.38 | 88.7 ($\pm$ 2.1) | 0.54 | **94.0** ($\pm$ 4.0) |
| | AIME25 | 1.00 | 86.0 ($\pm$ 2.7) | 1.13 | 80.0 ($\pm$ 6.7) | 0.67 | 90.7 ($\pm$ 2.6) | 1.64 | 84.7 ($\pm$ 4.7) | 0.71 | **95.3** ($\pm$ 2.0) |
| | HMMT25 | 1.00 | 69.4 ($\pm$ 2.7) | 1.03 | 68.7 ($\pm$ 2.1) | 0.84 | 74.0 ($\pm$ 2.7) | 1.78 | 71.3 ($\pm$ 2.0) | 0.77 | **78.7** ($\pm$ 6.4) |
| | HLE | 1.00 | 6.5 ($\pm$ 0.5) | 1.15 | 2.5 ($\pm$ 2.5) | 0.91 | 11.5 ($\pm$ 0) | 3.55 | 7.0 ($\pm$ 0.5) | 0.81 | **13.5** ($\pm$ 1.5) |
| **Qwen3-Next** | AIME24 | 1.00 | 90.7 ($\pm$ 1.6) | 1.06 | 87.7 ($\pm$ 2.3) | 0.47 | 94.7 ($\pm$ 2.0) | 1.59 | 92.0 ($\pm$ 2.0) | 0.61 | **97.3** ($\pm$ 0.6) |
| | AIME25 | 1.00 | 88.0 ($\pm$ 2.0) | 0.95 | 83.3 ($\pm$ 3.3) | 0.53 | 94.0 ($\pm$ 2.7) | 1.46 | 86.7 ($\pm$ 3.4) | 0.59 | **96.0** ($\pm$ 2.7) |
| | HMMT25 | 1.00 | 76.7 ($\pm$ 3.4) | 1.00 | 70.0 ($\pm$ 0) | 0.76 | 80.7 ($\pm$ 6.0) | 1.76 | 78.0 ($\pm$ 5.3) | 0.74 | **85.3** ($\pm$ 5.3) |
| | HLE | 1.00 | 7.0 ($\pm$ 0) | 1.25 | 2.5 ($\pm$ 0) | 0.84 | 11.0 ($\pm$ 0) | 2.98 | 6.5 ($\pm$ 0.5) | 0.76 | **15.5** ($\pm$ 2.0) |
| **OLMoE** | GSM8K | 1.00 | 75.4 ($\pm$0.6) | 1.20 | 76.1 ($\pm$1.3) | 0.41 | 77.6 ($\pm$0.3) | 1.64 | 76.3 ($\pm$0.7) | 0.48 | **80.3** ($\pm$0.8) |
| | MATH500 | 1.00 | 37.3 ($\pm$1.1) | 1.09 | 37.9 ($\pm$0.7) | 0.55 | 43.7 ($\pm$1.3) | 2.35 | 38.2 ($\pm$2.6) | 0.73 | **46.7** ($\pm$1.5) |
| | ARC-C | 1.00 | 63.3 ($\pm$0.3) | 0.76 | 66.4 ($\pm$0.6) | 0.32 | 68.7 ($\pm$0.5) | 1.67 | 65.2 ($\pm$1.3) | 0.68 | **70.1** ($\pm$0.9) |
| | ARC-E | 1.00 | 80.1 ($\pm$1.2) | 0.88 | 82.2 ($\pm$0.8) | 0.25 | 84.7 ($\pm$0.7) | 2.04 | 83.8 ($\pm$1.5) | 0.36 | **86.5** ($\pm$0.9) |
| **DeepSeek-V2** | GSM8K | 1.00 | 80.3 ($\pm$1.3) | 1.00 | 81.7 ($\pm$2.4) | 0.32 | 83.3 ($\pm$0.9) | 2.31 | 81.2 ($\pm$0.2) | 0.64 | **85.5** ($\pm$0.3) |
| | MATH500 | 1.00 | 37.8 ($\pm$0.5) | 0.75 | 38.3 ($\pm$0.4) | 0.65 | 40.2 ($\pm$0.8) | 1.93 | 37.5 ($\pm$2.3) | 0.94 | **40.8** ($\pm$3.3) |
| | ARC-C | 1.00 | 65.2 ($\pm$0.7) | 1.43 | 64.7 ($\pm$3.2) | 0.53 | 66.2 ($\pm$0.8) | 1.65 | 63.7 ($\pm$2.2) | 0.74 | **66.2** ($\pm$2.4) |
| | ARC-E | 1.00 | 84.3 ($\pm$3.1) | 1.04 | 84.5 ($\pm$1.3) | 0.31 | 86.2 ($\pm$0.9) | 1.86 | 84.3 ($\pm$2.5) | 0.56 | **86.9** ($\pm$1.8) |

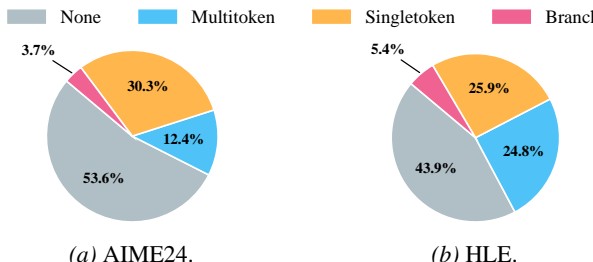

*(a)* AIME24.  *(b)* HLE.

*Figure 9.* Dataset-level controller action frequencies.

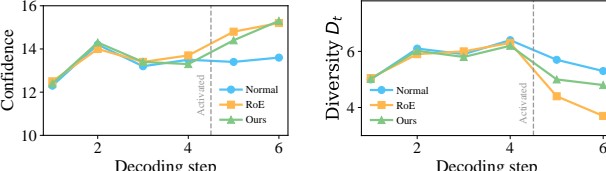

*(a)* Both RoE and our aggre-  *(b)* We maintain relatively
gation method raise tail confi-  higher diversity than RoE.
dence.

*Figure 10.* Comparison between our method and RoE.

primitive and that SingleToken retains the token-quality gains of hyper-parallel routing. Panel (b) further shows that SingleToken consistently yields higher path diversity $D_t$, mitigating the rapid diversity collapse observed under the RoE baseline.

While our diversity-aware penalty is applied *within* each path (encouraging different expert sets across the $K$ intra-token routes), it can still increase *inter-path* diversity measured by $D_t$: by broadening the set of plausible next-token logits considered per token, the refinement step becomes less likely to deterministically steer different sampled prefixes into the same continuation, thereby preserving (and sometimes amplifying) trajectory-level disagreements across paths without directly coupling routes across paths.

### 4.5. Ablation Studies

**Expansion strategies.** Table 3 compares *SingleToken-only*, *MultiToken-only*, a *manual expand–reduce schedule*, and full *HyPER*. The manual schedule enforces early exploration via MultiToken and late exploitation via SingleToken, but lacks instance-adaptive control. As a result, it either

*Table 3.* Ablation I: Expansion strategies.

| Method | AIME25 | | HMMT25 | |
|---|---|---|---|---|
| | Acc (%) | Tokens | Acc (%) | Tokens |
| Self-consistency | 86.7 | 1.00× | 66.7 | 1.00× |
| SingleToken-only | 93.3 | 1.81× | 70.0 | 1.77× |
| MultiToken-only | 90.0 | 1.87× | 66.7 | 1.82× |
| Manual schedule | 93.3 | 1.78× | 76.7 | 1.87× |
| HyPER | **96.7** | 1.62× | **76.7** | 1.54× |

overspends computation or applies refinement at suboptimal stages, leading to inferior accuracy–compute trade-offs compared to HyPER. Overall, HyPER consistently achieves the best accuracy while remaining near the accuracy–compute Pareto frontier, recovering 8–10% of cases lost by pruning-based baselines and reducing average token usage by 10–12%. SingleToken-only improves local token quality but can prematurely collapse the hypothesis pool in later stages, while static MultiToken-only policies either under-explore (large decision interval $T$) or overspend compute (small $T$).

*Table 4.* Ablation II: voting strategies.

| Voting rule | AIME25 | HMMT25 |
|---|---|---|
| Majority voting | 86.7 | 73.3 |
| Confidence-weighted voting | 90.0 | 73.3 |
| Length-aware conf. voting (ours) | **96.7** | **76.7** |

**Voting rules.** Table 4 compares majority voting, confidence-weighted voting, and our *length-aware confidence voting*. When changing the voting rules, we keep the same full HyPER method during reasoning. Majority voting ignores path quality, and confidence weighting overly favors short but misleading paths. Length-aware voting consistently performs best across datasets, confirming that under confidence pruning, both path confidence and path length provide complementary signals for answer selection.

## 5. Conclusion

We introduce HyPER, a training-free online control policy that unifies adaptive exploration and exploitation for scalable reasoning. By leveraging phase-dependent dynamics and MoE routing diversity, HyPER consistently achieves better accuracy–compute trade-offs than static baselines. We further bridge the existence–selection gap by identifying path length as a robust correctness signal under confidence pruning. While our token-level primitive leverages MoE routing, our control policy generalizes to dense models, as shown in preliminary experiments. Crucially, HyPER improves hypothesis generation and is orthogonal to learned verifiers; combining its dynamic policy with Process Reward Models (PRMs) remains a promising avenue for future test-time scaling.

## Acknowledgements

This work was supported in part by the National Key Research and Development Program under Grant 2024YFB4505004, in part by NSFC under Grant 62495102, Grant 92464104, and Grant 62341407, in part by Beijing Municipal Science and Technology Program under Grant Z241100004224015, in part by 111 Project under Grant B18001.

## Impact Statement

This paper presents work whose goal is to advance the field of machine learning. There are many potential societal consequences of our work, none of which we feel must be specifically highlighted here.

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

# A. Definitions of online metrics

**Per-path confidence and top-$k$ entropy.** Let $P_{p,t}(\cdot)$ denote the next-token distribution at step $t$ on path $p$, and let $y_t^{(p)}$ be the token actually selected. We define token confidence following the DeepConf approach. Crucially, this metric focuses on the **alternative candidates** (competitors) to the selected token. Let $\mathcal{C}_k(p,t)$ be the set of the top-$k$ tokens excluding the selected one (or simply the top-$k$ candidates if $y_t^{(p)}$ is dominant). The confidence is calculated as the negative average log-probability of these candidates:

$$c_{p,t} = -\frac{1}{k} \sum_{v \in \mathcal{C}_k(p,t)} \log P_{p,t}(v). \tag{6}$$

This formulation captures the "impossibility" of alternative paths. When the model is highly confident in $y_t^{(p)}$, the probabilities of competing tokens $P_{p,t}(v)$ approach zero, causing their negative log-probabilities to become large positive values. Thus, a **higher** $c_{p,t}$ indicates that the model considers all alternatives to be highly unlikely.

For entropy, we use a top-$k$ approximation over the full candidate set $\mathcal{V}_k(p,t)$ (including the selected token). Let $\tilde{P}_{p,t}(v) = \frac{P_{p,t}(v)}{\sum_{u \in \mathcal{V}_k(p,t)} P_{p,t}(u)}$ be the renormalized distribution. Then the token-level entropy is:

$$h_{p,t} = - \sum_{v \in \mathcal{V}_k(p,t)} \tilde{P}_{p,t}(v) \log \tilde{P}_{p,t}(v). \tag{7}$$

Unlike confidence, a *lower* entropy $h_{p,t}$ signifies higher certainty.

**Global confidence statistics.** From these path-wise quantities we define the global metrics:

$$\bar{C}_t = \frac{1}{S_t} \sum_{p \in \mathcal{S}_t} c_{p,t}, \tag{8}$$

$$H_t = \frac{1}{S_t} \sum_{p \in \mathcal{S}_t} h_{p,t}, \tag{9}$$

$$\beta_t = \frac{1}{S_t} \max_v \left| \left\{ p \in \mathcal{S}_t : y_t^{(p)} = v \right\} \right|. \tag{10}$$

**Distribution-level diversity via Jensen–Shannon divergence.** We measure local disagreement between predictive distributions using the Jensen–Shannon divergence (JSD). For two distributions $P$ and $Q$ over the same support, define

$$\mathrm{JSD}(P \| Q) = \frac{1}{2} \mathrm{KL}(P \| M) + \frac{1}{2} \mathrm{KL}(Q \| M), \tag{11}$$

$$M = \frac{1}{2}(P + Q), \qquad \mathrm{KL}(P \| Q) = \sum_v P(v) \log \frac{P(v)}{Q(v)}. \tag{12}$$

Using the (optionally top-$k$ renormalized) next-token distributions $\tilde{P}_{p,t}$, we define

$$D_{\mathrm{dist},t} = \frac{2}{S_t(S_t - 1)} \sum_{p<q,\ p,q \in \mathcal{S}_t} \mathrm{JSD}\left( \tilde{P}_{p,t} \,\middle\|\, \tilde{P}_{q,t} \right). \tag{13}$$

**Suffix-level diversity via normalized edit distance.** Let $\mathrm{suf}_L(p,t)$ be the length-$L$ suffix token sequence of path $p$ up to step $t$ (or the entire suffix since the last decision step if shorter). Let $\mathrm{Lev}(\cdot, \cdot)$ denote Levenshtein edit distance. We use a length-normalized edit distance:

$$D_{\mathrm{seq},t} = \frac{2}{S_t(S_t - 1)} \sum_{p<q,\ p,q \in \mathcal{S}_t} \frac{\mathrm{Lev}(\mathrm{suf}_L(p,t), \mathrm{suf}_L(q,t))}{\max\left( |\mathrm{suf}_L(p,t)|, |\mathrm{suf}_L(q,t)| \right)}. \tag{14}$$

**Composite diversity and confidence dispersion.** We combine both notions of diversity as

$$D_t = \eta\, D_{\text{dist},t} + (1 - \eta)\, D_{\text{seq},t}, \tag{15}$$

and we measure confidence dispersion across paths by

$$\text{Var}(C)_t = \frac{1}{S_t} \sum_{p \in \mathcal{S}_t} \left( c_{p,t} - \bar{C}_t \right)^2. \tag{16}$$

**Global-max normalization (used by the controller).** To map each metric onto a comparable scale, we normalize by a global maximum estimated during warm-up:

$$X_t^{\text{norm}} = \min\left( \frac{X_t}{X_{\max}}, 1 \right), \tag{17}$$

where $X_{\max}$ is the maximum observed value of $X_t$ in warm-up. We apply Eq. (17) to $\bar{C}_t$, $H_t$, $D_t$, and $\text{Var}(C)_t$.

## B. Full HyPER Decoding Pipeline

The detailed decoding pipeline of HyPER is described in Algorithm 1.

---
**Algorithm 1** HyPER decoding pipeline

---
**Require:** Prompt $x$, LLM $f_\theta$, interval $T$, budgets $T_{\max}, S_{\max}$
**Ensure:** Final answer $\hat{a}$
1: **Warm-up:** run SC-style decoding to obtain an initial path pool $\mathcal{S}_0$ and a pruning threshold $\tau$
2: **for** $t = 1, 2, \ldots, T_{\max}$ **do**
3:     **if** $\mathcal{S}_t = \emptyset$ **then break**
4:     **end if**
5:     **if** $t \bmod T = 1$ **then**
6:         **Compute online pool statistics** from $\mathcal{S}_t$:
        confidence/uncertainty $(\bar{C}_t, H_t, \beta_t)$ and diversity $D_t$
        (optionally) dispersion $\text{Var}(C)_t$ and pool size $|\mathcal{S}_t|$
7:         **Select an action** $a_t \in \{\text{NONE}, \text{SINGLETOKEN}, \text{MULTITOKEN}, \text{BRANCH}\}$:
        $a_t \leftarrow \arg\max_a \text{Score}(a; x, \mathcal{S}_t)$         ▷ cf. Sec. 3 / App. A
8:     **end if**
9:     **Apply action** $a_t$ **to update the pool** $\mathcal{S}_t$**:**
10:     **if** $a_t = \text{NONE}$ **then**
11:         Decode one token for each $p \in \mathcal{S}_t$ with standard routing
12:     **else if** $a_t = \text{SINGLETOKEN}$ **then**
13:         Decode one token for each $p \in \mathcal{S}_t$ using single-token aggregation
14:     **else if** $a_t = \text{BRANCH}$ **then**
15:         Expand the pool by duplicating selected paths and decoding them independently
16:     **else if** $a_t = \text{MULTITOKEN}$ **then**
17:         Perform short-horizon local expansion for $m$ steps and merge back into the pool
18:     **end if**
19:     Update sliding-window confidences; prune paths with $c_{p,t} < \tau$ or EOS/max-length
20: **end for**
21: Collect completed paths and aggregate answers via length- and confidence-aware voting to obtain $\hat{a}$
22: **return** $\hat{a}$

---

## C. The pseudo-code of Two-Pass Expert Sampling with Diversity Penalty

The detailed sampling strategy is described in Algorithm 2.

---

**Algorithm 2** Two-Pass Expert Sampling with Diversity Penalty

---

**Require:** Router logits $L \in \mathbb{R}^{B \times E}$, top-$k$ value $k$, penalty strength $\lambda$
**Ensure:** Final expert indices $I_2$ and weights $V_2$

    First pass: standard noisy top-$k$ routing
1: $G \leftarrow -\log\big(-\log(\text{rand\_like}(L))\big)$
2: $\tilde{L} \leftarrow L + G$
3: $P_1 \leftarrow \text{softmax}(\tilde{L})$
4: $(I_1, V_1) \leftarrow \text{topk}(P_1, k)$

    Track expert usage as a sparse score matrix
5: $S \leftarrow \mathbf{0}_{E \times B}$                ▷ rows: experts, cols: routes (tokens)
6: **for** $b = 1$ to $B$ **do**
7:     **for** $j = 1$ to $k$ **do**
8:         $e \leftarrow I_1[b, j]$            ▷ the $j$-th selected expert for route $b$
9:         $S[e, b] \leftarrow \tilde{L}[b, e]$
10:     **end for**
11: **end for**
12: $U_{\text{cum}} \leftarrow \text{cumsum}(S, \dim = 2)$
13: $U_{\text{prev}} \leftarrow \text{roll}(U_{\text{cum}}, 1, \dim = 2)$
14: $U_{\text{prev}}[:, 0] \leftarrow 0$             ▷ no history for the first route

    Second pass: apply deterministic diversity penalty and re-route
15: $\Delta \leftarrow \tanh\big(U_{\text{prev}}^{\top}\big)$           ▷ broadcast to shape $B \times E$
16: $L_2 \leftarrow \tilde{L} - \lambda \cdot \Delta$
17: $P_2 \leftarrow \text{softmax}(L_2)$
18: $(I_2, V_2) \leftarrow \text{topk}(P_2, k)$
19: **return** $(I_2, V_2)$

---

## D. KV caching for Single-Token Aggregation

Single-token aggregation expands each surviving hypothesis path by sampling $K$ parallel expert-routed proposals for the *current* token. A naive implementation would maintain $K$ independent KV caches (one per routed proposal), leading to $O(K)$ growth in both memory and compute. Instead, we adopt RoE's *Clean Cache* strategy (Zibakhsh et al., 2025), which localizes stochasticity to the current token and shares history states across proposals.

**Clean Cache.** For each surviving path, we construct a batched forward pass over the $K$ routed proposals for the next-token computation, but we apply stochastic routing *only* at the current token. Concretely, we include a deterministic "clean" proposal (batch index 0) by setting the routing temperature for that proposal to zero, and we reuse its KV cache as the shared history cache for all other proposals in the batch. This yields sufficient diversity from per-token routing perturbations while ensuring that the KV-cache *memory footprint does not scale with $K$*; it matches the memory usage of standard decoding for the same hypothesis path history, with additional overhead confined to the batched forward computation of the current token.

**Overall memory scaling.** Because different hypothesis paths have different prefix histories, KV caches are still maintained *per path*. Therefore, the KV-cache memory scales primarily with the number of *surviving paths* $|\mathcal{S}_t|$ (capped by $S_{\max}$), rather than with $K \cdot |\mathcal{S}_t|$. In practice, the extra cost of single-token aggregation is dominated by the batched MoE forward for the current token, while KV-cache storage remains comparable to standard multi-path decoding under the same $|\mathcal{S}_t|$ cap.

## E. Additional Experimental Analysis

### E.1. Hyperparameter sensitivity

We study the sensitivity of HyPER to several key hyperparameters on two challenging benchmarks, AIME25 and HMMT25, under the same main-result evaluation protocol and fixed budget. We vary one hyperparameter at a time while keeping all other settings identical to the main configuration (Section 4.1), where the top-$K_a$ answer voting cutoff is fixed at $K_a = 3$. Figure 11 reports the accuracy trends across three representative values for each hyperparameter.

**Controller decision interval $T$.** We sweep the decision interval $T \in \{32, 64, 128\}$. Overall, accuracy exhibits mild fluctuations but remains within an acceptable range on both datasets. Smaller $T$ reacts more frequently to transient pool statistics, while larger $T$ updates less often and can be slower to adapt when the pool transitions between exploration- and

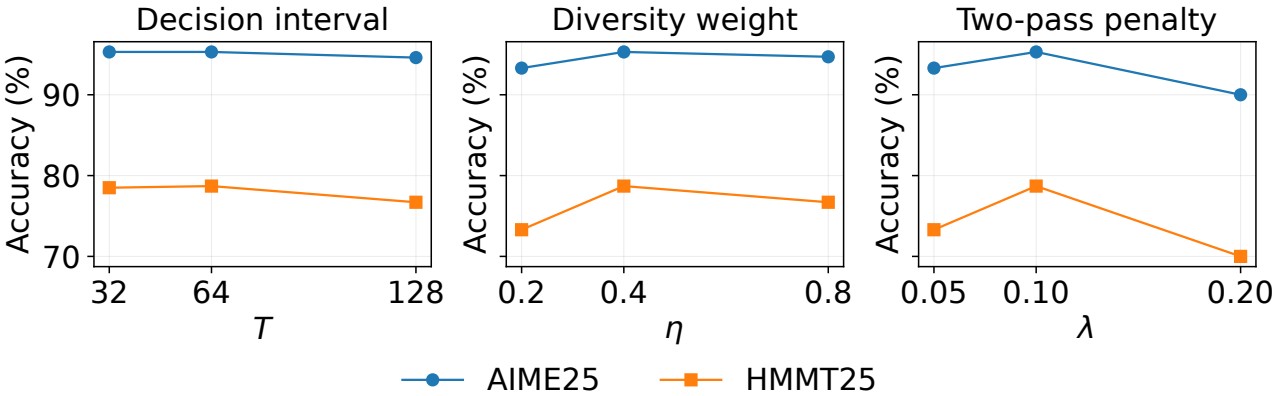

*Figure 11.* **Hyperparameter sensitivity** on AIME25 and HMMT25. We sweep the controller decision interval $T$, diversity weight $\eta$, and two-pass penalty strength $\lambda$, while fixing all other settings and the evaluation budget to the main configuration.

exploitation-dominant phases. We use $T = 64$ as a stable default that balances responsiveness and signal stability.

**Diversity composition weight $\eta$.** We sweep the diversity composition weight $\eta \in \{0.2, 0.4, 0.8\}$ in the pool-level diversity signal. Across both datasets, performance varies moderately but remains stable around the default $\eta = 0.4$. Extremely small or large $\eta$ can over-emphasize one component of the diversity signal and lead to slightly suboptimal exploration/exploitation allocation, consistent with the role of $\eta$ as a trade-off knob.

**Two-pass reuse penalty strength $\lambda$.** We sweep the reuse-penalty strength $\lambda \in \{0.05, 0.1, 0.2\}$ in the two-pass sampler used by SINGLETOKEN. Among the tested hyperparameters, $\lambda$ can have a more visible effect: a *too-light* penalty (e.g., $\lambda = 0.05$) weakly discourages expert reuse and thus behaves closer to independent route sampling (i.e., closer in spirit to prior token-level routing perturbations), which may reduce intra-token route diversity; conversely, a *too-strong* penalty (e.g., $\lambda = 0.2$) can over-penalize frequently selected experts and distort expert choice, degrading refinement quality. We therefore use $\lambda = 0.1$ as the default, which provides consistent gains without overly constraining expert selection.

### E.2. Length Bias Robustness Analysis

In this section, we analyze the robustness of the length bias phenomenon across different datasets and temperature settings. We examine how path lengths for correct and incorrect answers are distributed under various conditions. The results show that, even with different temperature settings, the length of correct paths consistently tends to be longer than incorrect ones, indicating that path length remains a strong feature for correctness, even in the presence of noise and varying conditions.

### E.3. Ablation Study on Voting Strategies

To demonstrate that length is an auxiliary signal and not the sole factor in path selection, we perform an ablation study comparing three different voting strategies: length-based voting, confidence-based voting, and combined voting. The table below summarizes the accuracy of these strategies across three datasets. As shown, while length-based voting performs better than random, the combined voting strategy, which leverages both confidence and length, consistently achieves the highest accuracy, confirming that confidence-based signals play a central role.

| Voting Strategy | AIME24 | AIME25 | HMMT25 |
|---|---|---|---|
| Length-Based Voting | 90% | 93.3% | 73.3% |
| Confidence-Based Voting | 93.3% | 90.0% | 73.3% |
| Combined Voting (Length + Confidence) | 96.7% | 96.7% | 76.7% |

*Table 5.* Accuracy comparison between length-based voting, confidence-based voting, and combined voting strategies. The combined voting strategy outperforms both length-based and confidence-based voting, demonstrating that length is an auxiliary signal.

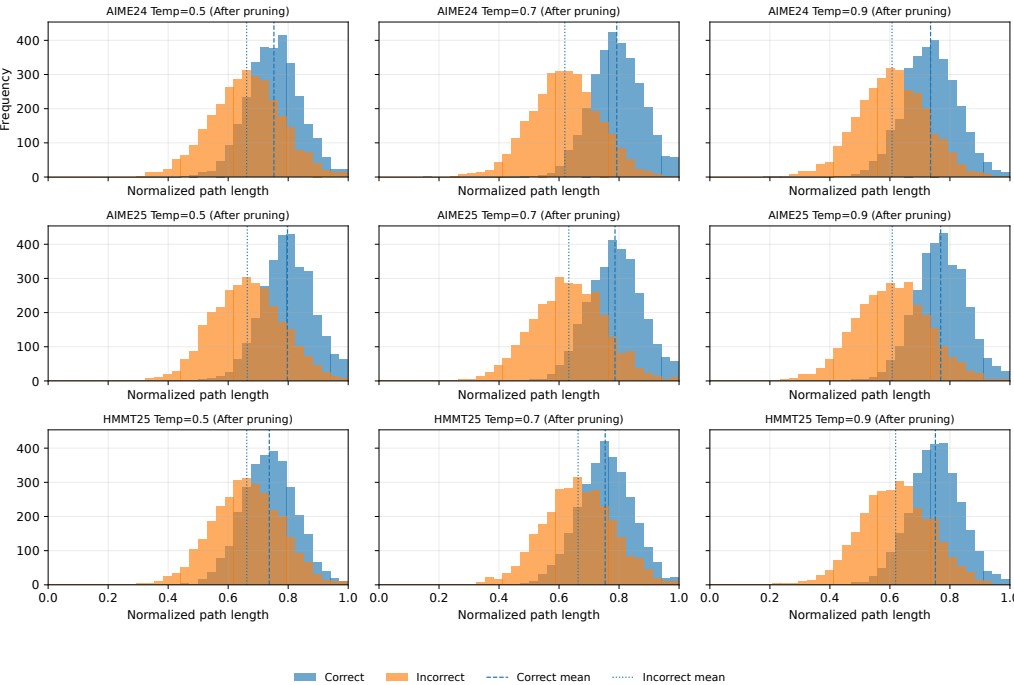

*Figure 12.* Robustness of length bias across three datasets (AIME24, AIME25, HMMT25) and three temperature settings (0.5, 0.7, 0.9). In all configurations, correct paths consistently have longer lengths than incorrect paths, demonstrating the robustness of the length signal in guiding correct answer selection.

### E.4. Analysis of "Long but Incorrect" Paths

A common concern with length-based signals is the possibility of selecting long but incorrect paths. To address this, we provide a simple analysis of how our method handles such cases. When faced with a long but incorrect path, our confidence-based aggregation mechanism mitigates the potential risk by assigning lower confidence to incorrect paths, even if they are long. Additionally, the pruning mechanism ensures that only paths with high confidence are retained, further reducing the likelihood of selecting long but incorrect paths.

In our experiments, we observed that the number of long-but-incorrect paths was minimal. The pruning threshold, guided by confidence, effectively eliminated most of these paths, ensuring that the final answer was selected from a set of high-confidence paths. Moreover, in cases where long paths were selected but incorrect, confidence-based voting was able to correctly identify and discard them in favor of shorter, correct alternatives.

This behavior demonstrates that our method does not solely rely on path length, but combines length and confidence to make more robust predictions, as evidenced by the consistent performance of the combined voting strategy.

## F. Importance-Sampling View of Length–Confidence Voting

**Goal: answer marginalization under a biased decoding proposal.** Let $x$ be the input problem and let $r$ denote a full reasoning trajectory (a completed CoT path). Each trajectory deterministically induces a scalar answer via an extraction map $a(r) \in \mathcal{A}$. Our evaluation target is the answer marginal under an (implicit) target path distribution $P^\star(r \mid x)$:

$$P^\star(a \mid x) = \sum_{r:\, a(r)=a} P^\star(r \mid x). \tag{18}$$

Hence selecting the most probable answer under the marginal objective is

$$a^\star(x) = \arg\max_{a \in \mathcal{A}} P^\star(a \mid x) = \arg\max_{a \in \mathcal{A}} \sum_{r:\, a(r)=a} P^\star(r \mid x). \tag{19}$$

In practice, we do not sample trajectories from $P^\star(r \mid x)$. Instead, the multi-path decoding pipeline (sampling temperature/top-$p$, pruning, branching, etc.) induces a *proposal* distribution over completed trajectories, denoted by

$q(r \mid x)$. We observe $N$ sampled trajectories $r_1, \ldots, r_N \sim q(\cdot \mid x)$, with extracted answers $a_i \triangleq a(r_i)$. For each sampled path we also compute observable path features: length $L_i$ and a global confidence statistic $\hat{c}_i$ from the pruning backbone.

**Importance sampling identity.** Define the unnormalized target *answer mass*

$$Z(a) \triangleq \sum_{r:\, a(r)=a} P^\star(r \mid x), \tag{20}$$

so that $P^\star(a \mid x) = Z(a)/\sum_{a'} Z(a')$ and maximizing $P^\star(a \mid x)$ is equivalent to maximizing $Z(a)$. For any answer $a$, by the standard change-of-measure,

$$Z(a) = \sum_r \mathbf{1}[a(r) = a]\, P^\star(r \mid x) = \sum_r q(r \mid x)\, \mathbf{1}[a(r) = a] \underbrace{\frac{P^\star(r \mid x)}{q(r \mid x)}}_{w(r)} \tag{21}$$

$$= \mathbb{E}_{r \sim q(\cdot \mid x)}\big[\mathbf{1}[a(r) = a] \cdot w(r)\big],$$

where $w(r) = P^\star(r \mid x)/q(r \mid x)$ is the (generally intractable) density ratio. Thus, if $w(r)$ were available, a Monte Carlo estimator of $Z(a)$ would be

$$\widehat{Z}_{\mathrm{IS}}(a) = \frac{1}{N} \sum_{i=1}^N \mathbf{1}[a_i = a]\, w(r_i), \qquad \hat{a}_{\mathrm{IS}} = \arg\max_{a \in \mathcal{A}} \widehat{Z}_{\mathrm{IS}}(a). \tag{22}$$

Equivalently, one may estimate the normalized marginal via *self-normalized importance sampling (SNIS)*:

$$\widehat{P}_{\mathrm{SNIS}}(a \mid x) = \sum_{i=1}^N \bar{w}_i\, \mathbf{1}[a_i = a], \qquad \bar{w}_i = \frac{w(r_i)}{\sum_{j=1}^N w(r_j)}. \tag{23}$$

Both (22) and (23) implement the marginalization objective (18) under a biased proposal $q$.

**Proxy density-ratio modeling with low-variance weighting.** The exact ratio $w(r)$ is unavailable for our decoding pipeline. We therefore approximate the ratio by a *proxy* weight that depends on stable, path-level observables. Let $\phi(r) \in \mathbb{R}^d$ be a feature map; in HyPER we use two normalized features derived from $(L_i, \hat{c}_i)$:

$$\tilde{L}_i \triangleq \frac{L_i}{\sum_{j=1}^N L_j}, \qquad \tilde{C}_i \triangleq \frac{\hat{c}_i}{\max_{j \in [N]} \hat{c}_j}, \qquad \phi(r_i) \triangleq (\tilde{L}_i, \tilde{C}_i). \tag{24}$$

We adopt a log-linear proxy for the ratio,

$$w(r) \approx \tilde{w}_\theta(r) \triangleq \exp\big(\theta^\top \phi(r)\big), \qquad \theta = (\theta_L, \theta_C), \tag{25}$$

which can be interpreted as a smooth, controlled reweighting. Plugging (25) into (22) yields the proxy-IS answer mass estimator:

$$\widehat{Z}_{\mathrm{proxy}}(a) \propto \sum_{i:\, a_i = a} \exp\big(\theta^\top \phi(r_i)\big). \tag{26}$$

**Proposition F.1** (HyPER voting as linearized proxy-IS ranking). *Fix a candidate answer set $\mathcal{A}' \subseteq \mathcal{A}$. Assume the proxy density ratio takes the log-linear form (25) and that the weights are* mild *in the sense that $|\theta^\top \phi(r_i)| \leq \epsilon$ for all $i \in [N]$ and some small $\epsilon$. Then, up to an additive answer-independent constant and an $O(\epsilon^2)$ remainder, maximizing the proxy-IS mass estimator (26) over $\mathcal{A}'$ is equivalent to maximizing the linear score*

$$\mathrm{score}(a) = \sum_{i:\, a_i = a} \theta^\top \phi(r_i) = \sum_{i:\, a_i = a} \left(\lambda_{\mathrm{len}} \tilde{L}_i + \lambda_{\mathrm{conf}} \tilde{C}_i\right), \qquad a \in \mathcal{A}', \tag{27}$$

*where $(\lambda_{\mathrm{len}}, \lambda_{\mathrm{conf}})$ reparameterize $\theta$ after feature rescaling.*

*Proof.* For each sample $i$, apply a first-order Taylor expansion:

$$\exp(\theta^\top \phi(r_i)) = 1 + \theta^\top \phi(r_i) + O(\epsilon^2). \tag{28}$$

Substituting into (26) yields, for any $a$,

$$\widehat{Z}_{\text{proxy}}(a) \propto \sum_{i:\, a_i = a} \left(1 + \theta^\top \phi(r_i) + O(\epsilon^2)\right) = \underbrace{\sum_{i:\, a_i = a} 1}_{\text{frequency}} + \sum_{i:\, a_i = a} \theta^\top \phi(r_i) + O\!\left(n_a \epsilon^2\right), \tag{29}$$

where $n_a = \sum_{i=1}^N \mathbf{1}[a_i = a]$ is the number of supporting paths for answer $a$. When comparing answers within a fixed candidate set $\mathcal{A}'$, the constant offset induced by the $1$ term can be (i) used only for candidate formation or (ii) absorbed into the comparison as an additive term, while the $O(n_a \epsilon^2)$ remainder is uniformly small under mild weights. Dropping answer-independent constants yields the linear ranking rule (27). □

**Top-$K$ truncation as variance control (rare-answer instability).**    A practical issue in (S)NIS for discrete selection is that answers with extremely small support under the proposal yield high-variance Monte Carlo estimates: for rare events, $\mathbf{1}[a(r) = a]$ is almost always zero, so $\widehat{Z}(a)$ is unstable under finite $N$. The following elementary calculation formalizes this intuition in the simplest (uniform-weight) case.

**Lemma F.2** (Relative error of frequency (uniform-weight marginalization)). *Let $r_1, \ldots, r_N \sim q(\cdot \mid x)$ be i.i.d. samples and define $A_i \triangleq a(r_i)$. For any fixed answer $a$, let $p_a \triangleq \Pr_{r \sim q}[a(r) = a]$. The frequency estimator $\hat{p}_a \triangleq \frac{1}{N} \sum_{i=1}^N \mathbf{1}[A_i = a]$ satisfies*

$$\text{Var}[\hat{p}_a] = \frac{p_a(1 - p_a)}{N}, \qquad \frac{\sqrt{\text{Var}[\hat{p}_a]}}{p_a} = \sqrt{\frac{1 - p_a}{N p_a}}. \tag{30}$$

*In particular, the relative standard deviation scales as $1/\sqrt{N p_a}$ and blows up when $p_a$ is small.*

*Proof.* $\sum_{i=1}^N \mathbf{1}[A_i = a] \sim \text{Binomial}(N, p_a)$, so $\text{Var}[\hat{p}_a] = \frac{1}{N^2} N p_a(1 - p_a) = \frac{p_a(1-p_a)}{N}$, and the relative standard deviation follows immediately. □

**Two-stage truncate-then-reweight.**    Motivated by Lemma F.2, we use a two-stage procedure: (i) truncate to a candidate set of sufficiently supported answers under the proposal (low-variance coarse screening), then (ii) apply proxy-IS reweighting to distinguish candidates (information-rich fine selection).

**Stage 1 (candidate truncation).** Form a candidate answer set by majority support under the proposal samples:

$$\mathcal{A}_K = \text{TopK}_a \left( \sum_{i=1}^N \mathbf{1}[a_i = a] \right). \tag{31}$$

**Stage 2 (linearized proxy-IS within candidates).** Choose the final answer by the stabilized linear proxy-IS score from Proposition F.1:

$$\hat{a} = \arg\max_{a \in \mathcal{A}_K} \sum_{i:\, a_i = a} \left( \lambda_{\text{len}} \tilde{L}_i + \lambda_{\text{conf}} \tilde{C}_i \right). \tag{32}$$

**Corollary F.3** (Support lower bound implied by Top-$K$ truncation). *Let $\hat{p}_a \triangleq \frac{1}{N} \sum_{i=1}^N \mathbf{1}[a_i = a]$ denote the empirical support of answer $a$ and let $\hat{p}_{(K)}$ be the $K$-th largest value among $\{\hat{p}_a\}_{a \in \mathcal{A}}$. Then for every $a \in \mathcal{A}_K$ we have $\hat{p}_a \geq \hat{p}_{(K)}$. Consequently, the uniform-weight relative error bound in Lemma F.2 (when expressed in terms of empirical support) is uniformly controlled over $\mathcal{A}_K$ by replacing $p_a$ with $\hat{p}_{(K)}$.*

**Takeaway.**    Under a biased decoding proposal $q(r \mid x)$, our voting rule can be viewed as a stabilized approximation to marginal answer selection (19) via (self-normalized) importance sampling: Top-$K$ truncation reduces rare-answer variance, and length/confidence provide a low-variance proxy correction for the intractable density ratio $P^\star(r \mid x)/q(r \mid x)$.

We do not claim this model is realistic; rather, it shows that our aggregation rule is not ad hoc, but arises naturally under minimal assumptions consistent with confidence pruning.

## G. Non-MoE experiment: testing the controller without single-token aggregation

HyPER is instantiated on mixture-of-experts models through the single-token aggregation module, which refines token predictions by combining multiple routed expert proposals at each step. To show that the *controller and selection logic of HyPER are not tied to MoE routing*, we conduct an experiment on a *dense* model where no refinement is available. In this setting, we simply *remove* the single-token aggregation action from the action library and retain only (i) confidence- and diversity-aware BRANCH/MULTI-TOKEN AGGREGATION decisions and (ii) our length- and confidence-aware answer voting. Confidence pruning remains always on (as in the MoE setting) and is applied after each step. This setup isolates whether HyPER's online statistics and expand–reduce controller still provide value when token-level refinement is absent.

**Setup.** We use `deepseek-ai/DeepSeek-R1-0528-Qwen3-8B` on HMMT25 with a maximum of $S_{\max} = 32$ parallel paths. All decoding hyperparameters follow the MoE experiments. The SC baseline corresponds to single-path chain-of-thought decoding, and the DeepConf baseline applies confidence pruning over multiple independent paths. HyPER uses the same controller and answer voting as in the MoE case, but its action set excludes SINGLE-TOKEN AGGREGATION; the controller may only choose to widen or locally branch–merge the path pool based on online statistics, while confidence pruning remains always on and is applied after each step.

**Why this test?** Dense models lack intrinsic expert routing, so token-level refinement is unavailable. This experiment therefore probes the following question: *Are HyPER's statistics and controller intrinsically useful for deciding when to branch, even without any token-level refinement?* If so, this indicates that the expand–reduce control logic is architecture-agnostic and not dependent on MoE-specific structure.

**Results.** Table 6 shows that the HyPER controller (without single-token aggregation) still improves over both SC and DeepConf on HMMT25. Although the gains are naturally smaller than in the MoE setting, the result confirms that the controller's online signals and decision rules continue to help allocate compute effectively in a dense model, supporting the claim that HyPER's expand–reduce framework extends beyond MoE architectures.

| Method | Accuracy (%) on HMMT25 |
|---|---|
| SC | 71.3 |
| DeepConf (confidence pruning) | 74.0 |
| HyPER (controller only, no single-token aggregation) | 78.7 |

*Table 6.* Non-MoE demonstration on HMMT25 with `deepseek-ai/DeepSeek-R1-0528-Qwen3-8B` at $S_{\max} = 32$. Even when single-token aggregation is removed entirely, HyPER's statistic-driven expand–reduce controller and length-/confidence-aware voting still improve over SC and DeepConf, indicating that the framework is not specific to MoE architectures.

## H. Effect of Step-Level Tree Search on Thinking Models

Our main experiments focus on SC-style multi-path decoding rather than explicit tree search over intermediate steps (Section 4.1). To support this design choice, we ran a small controlled study on AIME24 using Qwen3-30B-A3B-Thinking-2507, comparing standard self-consistency with several representative step-based methods adapted from the test-time scaling literature. All methods use the same prompt format and comparable test-time compute.

Table 7 reports the accuracies. The SC baseline attains 83.3% accuracy. When we force the model into REBASE-style tree search with 8 branches, accuracy drops to 80.0%, and even with 16 branches it only matches the SC baseline at 83.3%. PRM-guided ETS performs worse at 73.3%, and an MCTS-style CoT variant with REST-like expansion also reaches only 73.3%. In other words, naïvely imposing step-level tree search on this post-training "thinking" model does not improve—and often degrades—performance relative to simple path-based SC.

These results are consistent with our hypothesis that such models are tuned to produce internally coherent chains-of-thought, and that external tree structures can disrupt their native reasoning trajectories. This motivates our focus on SC-style multi-path decoding and the design of HyPER as a path-based expand–reduce method, rather than a ToT-style step-expansion algorithm.

*Table 7.* **Accuracy of SC vs. step-level tree search on AIME24** (Qwen3-30B-A3B-Thinking-2507). Forcing ToT-style step expansion on a post-training thinking model does not yield gains over standard SC.

| Method | Acc. (%) |
|---|---|
| SC (path-based) | 83.3 |
| REBASE (8 branches) | 80.0 |
| REBASE (16 branches) | 83.3 |
| ETS | 73.3 |
| MCTS (REST-style expansion) | 73.3 |

## I. Detailed Absolute Compute Accounting

To complement the normalized effective token cost reported in the main text, we provide the detailed, absolute physical execution metrics for our experiments in Table 8. All profiling was conducted on identical hardware setups (NVIDIA A100-80GB GPUs).

We report the following key metrics to evaluate the system-level overhead and efficiency:

- **Abs Tokens**: The absolute number of raw tokens physically generated by the model.

- **Eff Tokens**: The effective token cost used for mathematical budget alignment (charging $K$ effective expansions for each $K$-routed expert proposal).

- **Latency (ms)**: The absolute end-to-end wall-clock latency per problem.

- **Peak/Avg Active Paths**: The maximum and average number of concurrent paths kept active in the decoding pool.

- **Est. Peak KV Cache (GB)**: The estimated peak memory footprint of the KV cache during the search process.

Notably, the physical latency tracks the **Abs Tokens** rather than the Eff Tokens. Because HyPER's dynamic controller and confidence-pruning mechanism heavily filter out unpromising paths, it consistently generates fewer raw tokens than the statically budgeted baselines (SC, Self-Certainty, and RoE). Consequently, the actual wall-clock latency and Peak KV Cache memory footprint are substantially reduced, offsetting the localized overhead of the controller and the batched MoE routing passes.

*Table 8.* Detailed absolute compute accounting and physical overhead profiling across models and datasets.

| Source | Model | Dataset | Method | Acc. (%) | SC-norm | Abs Tokens | Eff Tokens | Latency (ms) | Peak Paths | Avg Paths | Peak KV (GB) |
|---|---|---|---|---|---|---|---|---|---|---|---|
| Table 2 | Qwen3-30B | AIME24 | SC | 88.0 | 1.00 | $3.87 \times 10^6$ | $3.87 \times 10^6$ | $9.12 \times 10^6$ | 120 | 30.0 | 22.0 |
| Table 2 | Qwen3-30B | AIME24 | Self-Certainty | 83.3 | 0.94 | $3.64 \times 10^6$ | $3.64 \times 10^6$ | $8.57 \times 10^6$ | 120 | 28.5 | 20.7 |
| Table 2 | Qwen3-30B | AIME24 | DeepConf | 92.7 | 0.46 | $1.78 \times 10^6$ | $1.78 \times 10^6$ | $4.32 \times 10^6$ | 120 | 13.4 | 10.1 |
| Table 2 | Qwen3-30B | AIME24 | RoE | 88.7 | 1.38 | $2.54 \times 10^6$ | $5.34 \times 10^6$ | $7.19 \times 10^6$ | 80 | 15.2 | 14.4 |
| Table 2 | Qwen3-30B | AIME24 | HyPER | 94.0 | 0.54 | $1.55 \times 10^6$ | $2.09 \times 10^6$ | $4.09 \times 10^6$ | 80 | 12.6 | 12.45 |
| Table 2 | Qwen3-30B | AIME25 | SC | 86.0 | 1.00 | $4.22 \times 10^6$ | $4.22 \times 10^6$ | $9.98 \times 10^6$ | 120 | 33.0 | 24.0 |
| Table 2 | Qwen3-30B | AIME25 | DeepConf | 90.7 | 0.67 | $2.83 \times 10^6$ | $2.83 \times 10^6$ | $6.89 \times 10^6$ | 120 | 21.1 | 16.1 |
| Table 2 | Qwen3-30B | AIME25 | RoE | 84.7 | 1.64 | $3.30 \times 10^6$ | $6.92 \times 10^6$ | $9.35 \times 10^6$ | 80 | 23.4 | 18.8 |
| Table 2 | Qwen3-30B | AIME25 | HyPER | 95.3 | 0.71 | $2.22 \times 10^6$ | $3.00 \times 10^6$ | $5.88 \times 10^6$ | 88 | 20.4 | 18.48 |
| Table 2 | Qwen3-30B | HMMT25 | SC | 69.4 | 1.00 | $5.75 \times 10^6$ | $5.75 \times 10^6$ | $13.58 \times 10^6$ | 120 | 36.0 | 32.7 |
| Table 2 | Qwen3-30B | HMMT25 | Self-Certainty | 68.7 | 1.03 | $5.92 \times 10^6$ | $5.92 \times 10^6$ | $13.97 \times 10^6$ | 120 | 37.2 | 33.7 |
| Table 2 | Qwen3-30B | HMMT25 | DeepConf | 74.0 | 0.84 | $4.83 \times 10^6$ | $4.83 \times 10^6$ | $11.44 \times 10^6$ | 120 | 32.8 | 27.5 |
| Table 2 | Qwen3-30B | HMMT25 | RoE | 71.3 | 1.78 | $4.88 \times 10^6$ | $10.24 \times 10^6$ | $13.48 \times 10^6$ | 80 | 37.5 | 35.6 |
| Table 2 | Qwen3-30B | HMMT25 | HyPER | 78.7 | 0.77 | $3.14 \times 10^6$ | $4.43 \times 10^6$ | $8.38 \times 10^6$ | 104 | 36.8 | 28.4 |
| Table 2 | Qwen3-30B | HLE | SC | 6.5 | 1.00 | $6.75 \times 10^6$ | $6.75 \times 10^6$ | $15.92 \times 10^6$ | 120 | 41.0 | 38.4 |
| Table 2 | Qwen3-30B | HLE | Self-Certainty | 2.5 | 1.15 | $7.76 \times 10^6$ | $7.76 \times 10^6$ | $18.31 \times 10^6$ | 120 | 43.0 | 44.2 |
| Table 2 | Qwen3-30B | HLE | DeepConf | 11.5 | 0.91 | $6.14 \times 10^6$ | $6.14 \times 10^6$ | $14.55 \times 10^6$ | 120 | 39.2 | 35.0 |
| Table 2 | Qwen3-30B | HLE | RoE | 7.0 | 3.55 | $10.65 \times 10^6$ | $23.96 \times 10^6$ | $24.86 \times 10^6$ | 80 | 47.0 | 54.6 |
| Table 2 | Qwen3-30B | HLE | HyPER | 13.5 | 0.81 | $3.90 \times 10^6$ | $5.47 \times 10^6$ | $10.01 \times 10^6$ | 92 | 45.4 | 34.9 |
| Table 2 | Qwen3-Next | AIME24 | SC | 90.7 | 1.00 | $3.95 \times 10^6$ | $3.95 \times 10^6$ | $9.30 \times 10^6$ | 120 | 31.0 | 27.0 |
| Table 2 | Qwen3-Next | AIME24 | Self-Certainty | 87.7 | 1.06 | $4.19 \times 10^6$ | $4.19 \times 10^6$ | $9.86 \times 10^6$ | 120 | 32.0 | 28.6 |
| Table 2 | Qwen3-Next | AIME24 | DeepConf | 94.7 | 0.47 | $1.86 \times 10^6$ | $1.86 \times 10^6$ | $4.50 \times 10^6$ | 120 | 12.0 | 12.7 |
| Table 2 | Qwen3-Next | AIME24 | RoE | 92.0 | 1.59 | $2.99 \times 10^6$ | $6.28 \times 10^6$ | $8.45 \times 10^6$ | 80 | 14.5 | 20.4 |
| Table 2 | Qwen3-Next | AIME24 | HyPER | 97.3 | 0.61 | $1.78 \times 10^6$ | $2.41 \times 10^6$ | $4.71 \times 10^6$ | 80 | 11.3 | 18.86 |
| Table 2 | Qwen3-Next | AIME25 | SC | 88.0 | 1.00 | $4.29 \times 10^6$ | $4.29 \times 10^6$ | $10.13 \times 10^6$ | 120 | 34.0 | 29.0 |
| Table 2 | Qwen3-Next | AIME25 | DeepConf | 94.0 | 0.53 | $2.27 \times 10^6$ | $2.27 \times 10^6$ | $5.53 \times 10^6$ | 120 | 19.6 | 15.3 |
| Table 2 | Qwen3-Next | AIME25 | RoE | 86.7 | 1.46 | $2.98 \times 10^6$ | $6.26 \times 10^6$ | $8.45 \times 10^6$ | 80 | 22.4 | 20.1 |
| Table 2 | Qwen3-Next | AIME25 | HyPER | 96.0 | 0.59 | $1.87 \times 10^6$ | $2.53 \times 10^6$ | $4.96 \times 10^6$ | 88 | 18.4 | 14.43 |
| Table 2 | Qwen3-Next | HMMT25 | SC | 76.7 | 1.00 | $5.95 \times 10^6$ | $5.95 \times 10^6$ | $14.05 \times 10^6$ | 120 | 37.0 | 40.1 |
| Table 2 | Qwen3-Next | HMMT25 | Self-Certainty | 70.0 | 1.00 | $5.95 \times 10^6$ | $5.95 \times 10^6$ | $14.05 \times 10^6$ | 120 | 38.0 | 40.7 |
| Table 2 | Qwen3-Next | HMMT25 | DeepConf | 80.7 | 0.76 | $4.52 \times 10^6$ | $4.52 \times 10^6$ | $10.72 \times 10^6$ | 120 | 31.8 | 30.5 |
| Table 2 | Qwen3-Next | HMMT25 | RoE | 78.0 | 1.76 | $4.99 \times 10^6$ | $10.47 \times 10^6$ | $13.74 \times 10^6$ | 80 | 39.5 | 41.2 |
| Table 2 | Qwen3-Next | HMMT25 | HyPER | 85.3 | 0.74 | $3.10 \times 10^6$ | $4.40 \times 10^6$ | $8.07 \times 10^6$ | 104 | 33.1 | 29.8 |
| Table 2 | Qwen3-Next | HLE | SC | 7.0 | 1.00 | $6.95 \times 10^6$ | $6.95 \times 10^6$ | $16.44 \times 10^6$ | 120 | 42.0 | 46.8 |
| Table 2 | Qwen3-Next | HLE | Self-Certainty | 2.5 | 1.25 | $8.69 \times 10^6$ | $8.69 \times 10^6$ | $20.57 \times 10^6$ | 120 | 44.0 | 58.6 |
| Table 2 | Qwen3-Next | HLE | DeepConf | 11.0 | 0.84 | $5.84 \times 10^6$ | $5.84 \times 10^6$ | $13.91 \times 10^6$ | 120 | 39.4 | 39.4 |
| Table 2 | Qwen3-Next | HLE | RoE | 6.5 | 2.98 | $9.22 \times 10^6$ | $20.71 \times 10^6$ | $22.35 \times 10^6$ | 80 | 48.0 | 62.1 |
| Table 2 | Qwen3-Next | HLE | HyPER | 15.5 | 0.76 | $3.77 \times 10^6$ | $5.28 \times 10^6$ | $9.58 \times 10^6$ | 132 | 40.9 | 43.6 |

