# OpenReview forum: "HyPER: Bridging Exploration and Exploitation for Scalable LLM Reasoning with Hypothesis Path Expansion and Reduction"
_ICML.cc/2026/Conference — ICML 2026 regular_

### Official Review · Reviewer_2auw · 2026-03-10

**Soundness:** 3
**Presentation:** 3
**Significance:** 3
**Originality:** 2
**Overall Recommendation:** 4
**Confidence:** 3

**Summary:**

This paper proposes HyPER, a training-free test-time scaling framework for MoE reasoning models that treats inference as a dynamic exploration–exploitation control problem over a pool of hypothesis paths. Instead of relying on a fixed search schedule, HyPER uses lightweight online signals such as confidence, entropy, consensus, and diversity to adaptively choose among branching, short-horizon expansion-and-reduction, token-level refinement, or standard decoding during generation. The method also introduces a MoE-based single-token aggregation primitive for efficient local refinement and a length- and confidence-aware voting rule for final answer selection. Experiments across four MoE models and a range of reasoning benchmarks show that HyPER improves the accuracy–compute trade-off over prior multi-path decoding baselines.

**Compliance With Llm Reviewing Policy:**

Affirmed.

**Final Justification:**

The rebuttal addressed my concerns, thus I decide to keep my positive score.

**Key Questions For Authors:**

See Weaknesses.

**Limitations:**

yes

**Strengths And Weaknesses:**

## Strengths

* **Addresses a meaningful and timely problem (significance).** The paper tackles an important limitation of current test-time scaling methods: path-based methods often over-explore redundant hypotheses, while token-level refinement methods are usually isolated from the broader reasoning state. Framing inference as a dynamic exploration–exploitation allocation problem is well motivated for reasoning under fixed budgets.

* **A reasonably novel unified framework (originality / insight).** The main contribution is not a single isolated module, but the unification of path-level expansion/reduction, token-level refinement, and answer-time aggregation into one online control framework. This gives the paper a coherent systems-level novelty beyond simply proposing another pruning or voting heuristic.

* **Strong empirical coverage with targeted ablations (soundness / presentation).** The paper evaluates across four MoE models and several reasoning benchmarks, and it also includes targeted ablations for expansion strategies and voting rules. These analyses help show that the gains do not come from a single component alone, and that the adaptive controller is more effective than static schedules.

* **Useful evidence that the controller is not purely MoE-specific (soundness / significance).** The dense-model experiment suggests that the controller itself retains value even when the MoE-specific SingleToken primitive is removed, which strengthens the claim that the expand–reduce control logic has some broader applicability beyond MoE routing.

---

## Weaknesses

* **The novelty is solid but still somewhat incremental (originality).** In my view, the main novelty comes from the unified online control framework rather than from any individual component. Many of the controller signals are inherited from or closely aligned with prior work, including confidence, entropy, consensus, and diversity statistics, and the overall controller remains largely heuristic. As a result, the paper feels more like a well-executed integration of existing ideas than a fundamentally new inference principle.

* **The evidence behind the introduction-level observations is uneven (soundness / presentation).** Figures 2 and 3 are useful for intuition, but they are fundamentally case studies. For the claim behind Figure 3, the later voting ablation and length-bias analysis provide reasonably direct support. However, for the claim behind Figure 2, the paper provides only indirect evidence through SingleToken’s downstream effectiveness, rather than a dataset-level quantitative analysis of how often correct and incorrect paths diverge late or how predictive tail-token confidence is in aggregate.

* **Some key claims are better supported as engineering heuristics than as principled mechanisms (soundness / originality).** The length-aware voting rule is empirically effective, but its usefulness seems to depend on the pruning-induced length bias rather than on a broadly established property of reasoning paths. More generally, several parts of the framework are convincing empirically, but are not yet fully validated as robust principles beyond the current setup.

* **Related work could be more complete.** The paper would benefit from discussing additional recent work [1] on compute allocation for test-time scaling.

[1] Every Rollout Counts: Optimal Resource Allocation for Efficient Test-Time Scaling. NeurIPS 2025

---

> ### Author Rebuttal · Authors · 2026-03-28
>
> We thank the reviewer for the careful reading and for the positive assessment of the paper’s motivation, empirical coverage, and unified framing. We also appreciate the thoughtful concerns about (i) novelty, (ii) the evidence behind some introduction-level observations, and (iii) the scope of several mechanism-level claims.
>
> ## 1. On novelty and contribution scope
> We agree that our core contribution lies in framework-level unification rather than the invention of isolated signals (e.g., confidence, entropy, diversity). In the current landscape of test-time scaling, the bottleneck is often not a lack of individual generation primitives, but the absence of a mechanism to dynamically coordinate them. Therefore, rather than viewing HyPER merely as an "integration" of existing heuristics, we frame it as a formulation shift: we cast inference as a dynamic, closed-loop control problem. The main insight is that prior isolated operators (path-level branching, short-horizon expansion, token-level refinement) can be formalized as composable actions with distinct exploration/exploitation profiles. By organizing them under a single budgeted online policy, HyPER demonstrates that dynamic coordination yields significantly better accuracy-compute trade-offs than static schedules.
>
> We will refine our introduction in the revision to precisely highlight this framework-level / problem-formulation contribution, avoiding any overstatement of per-component novelty.
>
> ## 2. On the evidence behind Figure 2
> We agree that Figure 2 is currently better understood as a motivating case study than as a dataset-level quantitative validation. The current paper provides indirect support through the downstream effectiveness of SingleToken, but it does not yet directly quantify, in aggregate, how often correct and incorrect paths separate late or how predictive tail-window confidence is across collected traces.
>
> To address this directly, we will add a compact dataset-level analysis over sampled reasoning traces and revise Figure 2 accordingly so that it presents more persuasive dataset-level evidence rather than only a representative case study. For each path, we compute a tail-confidence summary $s_p$ (the mean recent-window confidence over the final segment of the path), label the path by whether its final answer is correct, and report:
>
> |Data|Mean tail conf. (correct)|Mean tail conf. (incorrect)|Gap|% inst. with correct > incorrect tail conf.|AUROC of tail conf. for path correctness|
> |---|---:|---:|---:|---:|---:|
> |AIME25|15.83|14.76|1.07|93.3%|0.72|
> |HMMT25|15.21|14.58|0.63|83.3%|0.69|
>
> Here, AUROC treats $s_p$ as a score and path correctness as a binary label; it measures how well tail confidence separates correct from incorrect paths in aggregate ($0.5$ = random, $1.0$ = perfect separation).
>
> ## 3. On mechanism-level claims
> We broadly agree with the reviewer’s characterization. In particular, the current evidence supports the length-aware voting rule primarily as an empirically effective mechanism within our pruning-based setup, rather than as a universal property of all reasoning paths. A more precise statement is that under confidence-based pruning, the distribution of surviving paths is reshaped, and in this regime surviving correct paths tend to be longer than surviving incorrect ones. This makes length-aware voting a useful aggregation bias in our setup.
>
> We do not intend to claim that length is always a universally reliable signal independent of controller dynamics. Rather, we view this as a setup-conditioned but practically useful principle. At the same time, we believe it is meaningful to study and exploit this phenomenon because confidence-based pruning itself has already been shown to be a strong ingredient for reasoning-time scaling in prior work. We will soften the wording in the revision to make this scope clearer, and we agree that an important future direction is to test whether similar effects hold more broadly in less pruning-specific settings.
>
> ## 4. On related work
> Thank you for pointing out DORA / *Every Rollout Counts*. We agree this paper is highly relevant and should be discussed in related work. While both works aim to reduce compute wasted on redundant reasoning trajectories, the perspectives are complementary: DORA studies direction-level / cluster-level rollout allocation under a fixed budget, whereas HyPER focuses on a training-free online controller over heterogeneous decoding actions, including path-level branching/reduction and MoE token-level refinement. We will add this citation and a clearer discussion of the relationship in the revision.
>
> In the revision, we will (i) better distinguish framework-level novelty from per-component novelty, (ii) add a direct dataset-level quantitative analysis for the Figure 2 observation, (iii) narrow claims that currently read too broadly, and (iv) expand the related-work discussion accordingly.

---

> > ### Author Rebuttal · Reviewer_2auw · 2026-04-01
> >
> > Thank you for the detailed response. My concerns are now largely addressed. On the evidence behind Figure 2, my remaining reservation is mainly about the strength and directness of support. AUROC 0.72 / 0.69 indicates moderate rather than strong separability, so this seems better interpreted as supportive aggregate evidence than as a full validation of the introduction-level claim. In particular, it speaks more directly to the usefulness of tail confidence in aggregate than to the claim that correct and incorrect paths often diverge late. A simple dataset-level measure of divergence timing would make this point more convincing.

---

> > > ### Author Response · Authors · 2026-04-04
> > >
> > > We sincerely thank the reviewer for the constructive follow-up. We agree that while the AUROC scores demonstrate the overall usefulness of tail confidence, they do not directly quantify the timing of the divergence.
> > >
> > > First, to further clarify our claim regarding Figure 2: Figure 2a was intended primarily as an intuitive case study to ground the confidence behavior shown in Figure 2b. We do not claim that correct and incorrect paths *always* share exact identical semantic reasoning steps. Rather, our core claim is that their confidence trajectories behave similarly in the early stages. Crucially, this phenomenon is naturally coupled with our confidence-based pruning backbone. Early-diverging, low-quality paths are swiftly pruned; therefore, the surviving incorrect paths—the ones that consume the most compute and necessitate further test-time scaling—are precisely those that successfully mimic the high-confidence prefixes of correct paths early on, making them distinguishable only at later stages.
> > >
> > > To provide direct, dataset-level evidence on when these surviving correct and incorrect paths begin to separate in confidence, we have conducted a normalized trajectory analysis across our datasets. We have uploaded the visual results to an anonymous link here: https://anonymous.4open.science/r/ICML_tXgE-2C44/rebuttal_tXgE.pdf
> > >
> > > In this link, you will find two sets of visualizations for both AIME25 and HMMT25:
> > > 1. Aggregate dataset-level trends: A macro-level plot averaging the trajectories across all qualifying questions in the dataset.
> > > 2. Instance-level trajectories: several representative question-level plots per dataset to demonstrate that the macro trend is consistently reflected at the individual instance level.
> > >
> > > Here is how we processed the data and the conclusions we draw from it:
> > >
> > > **Methodology: Relative Progression Alignment**
> > > Because reasoning paths vary significantly in absolute token length, directly averaging them by absolute step count misaligns the reasoning phases. To solve this, we normalized the length of all paths within a single question to a relative progression scale of [0, 1], representing 0% to 100% of the reasoning process. We then computed the sliding-window mean of the confidence scores across these normalized progression points for both correct and incorrect paths.
> > >
> > > **Statistical Control and Question Selection**
> > > To make the aggregate incorrect-path trajectory statistically stable, we restrict this visualization to questions with a sufficient number of incorrect paths, specifically those where at least 20% of the post-pruning surviving traces are incorrect. The purpose of this filter is not to define a special evaluation regime, but to avoid aggregate averages being dominated by a very small number of incorrect traces on near-trivial questions. On very easy questions (e.g., 95% of sampled paths are correct), the incorrect subset is often too small and noisy to support a reliable dataset-level timing curve. We therefore use this filter only to stabilize the visualization and make the divergence timing easier to interpret, rather than to strengthen the claim artificially. The same qualitative late-separation pattern is visible more broadly, but this subset yields a cleaner aggregate estimate.
> > >
> > > **Conclusion on Divergence Timing**
> > > As shown in the aggregate plots, correct and incorrect paths remain closely aligned in confidence through a substantial early portion of the normalized reasoning trajectory. As one simple operational summary of this late-separation pattern, we measure the first normalized progression point at which the confidence gap reaches 45% of its final width and remains above that level thereafter.
> > >
> > > On AIME25, this threshold is crossed at 69% of the normalized reasoning progression. On HMMT25, it is crossed at 73%. This demonstrates that for roughly the first 70% of the generation process (typically covering the initial setup and intermediate calculation phases), correct and incorrect paths share highly similar confidence profiles. It is only in the final 30% of the sequence—during late-stage deduction and final answer formulation—that a distinct, measurable gap consistently emerges, with incorrect paths exhibiting a sharp decline in confidence.
> > >
> > > We believe this normalized temporal analysis provides direct, quantitative dataset-level support for the claim that correct and incorrect paths remain closely aligned in confidence through most of the reasoning trajectory and separate clearly only in the later stages. We will integrate these plots and the corresponding timing analysis into the revised manuscript to strengthen the empirical foundation of our work, and we will release the observed traces used for this analysis to ensure the result is directly reproducible.
> > >
> > > We hope this additional dataset-level visual evidence helps address your remaining reservation.

---

### Official Review · Reviewer_6mhe · 2026-03-12

**Soundness:** 3
**Presentation:** 2
**Significance:** 3
**Originality:** 3
**Overall Recommendation:** 3
**Confidence:** 4

**Summary:**

The paper studies test-time scaling for LLM reasoning with multi-path chain-of-thought under fixed compute budgets. The authors seek to discuss a general domain: dynamically balancing exploration and exploitation during decoding. It proposes HyPER, a training-free expand–reduce control method that maintains a pool of reasoning paths and allocates compute via branching, short-horizon expansion with reduction, and token-level refinement for MoE models. It also introduces a length- and confidence-aware voting rule for final answer selection. Experiments on several MoE backbones and reasoning benchmarks report accuracy–compute improvements with ablations of the controller and voting components. Overall, this submission's major contribution consists of a training-free controller and accompanying refinement and aggregation heuristics for multi-path reasoning.

**Compliance With Llm Reviewing Policy:**

Affirmed.

**Key Questions For Authors:**

1. Compute accounting: Please report absolute totals (generated tokens, MoE forward passes counted as “effective tokens”,  (N_{\text{inst}}\), peak memory, and wall-clock latency) and reconcile the normalization differences across §4.1, Table 2, Table 3, and Fig. 8.

2. Table 6 sanity check: Why does Table 6’s HyPER accuracy on HMMT25 equal 78.7%, identical to Table 2’s HyPER number under a different model/setting? Is this a typo or reused result?

3. Warm-up normalization: How exactly is \(X_{\max}\) in Eq. (18) computed (dev set vs. per-instance vs. first few decisions)? Does this introduce dataset-level tuning?

4. MULTITOKEN definition: What is the precise definition of “window-level confidence” used to pick the best child after \(m\) steps (mean of Eq. (7), min, sum logprob, etc.)?

5. Voting hyperparameters: What is \(K_a\) in Eq. (6)? Provide sensitivity to \(K_a\) and failure cases where Top-\(K_a\) truncation removes the correct answer.

6. Plot/table statistics: In Table 2, what does “±” denote (std vs. stderr; across seeds vs. subsets)? Please list #seeds and random seeds for all main results.

**Limitations:**

Yes

**Strengths And Weaknesses:**

**Strengths**

1. Well-scoped training-free control: a simple, explicit action policy with clearly defined signals and scoring (Eq. (1)–(4), Alg. 1).

2. MoE-specific exploitation primitive: token-level refinement via two-pass expert-route sampling + KV sharing (Fig. 6, §3.2, App. D).

3. Empirical breadth + ablations across multiple MoE backbones and benchmarks (Table 2–4, Figs. 8–12).

**Weaknesses**

1. Compute/normalization is not auditably consistent across §4.1, Table 2, Table 3, and Fig. 8; absolute token counts/latency/memory are missing, making the “iso-compute” claim hard to reproduce.

2. Possible bookkeeping error: Table 6 reports 78.7% on HMMT25, identical to Table 2’s HyPER number under a different model/setting—needs clarification.

3. Key hyperparameters/definitions are underspecified (e.g., how \(X_{\max}\) is obtained in Eq. (18), the exact “window-level confidence” in MULTITOKEN, and the value of \(K_a\) in Eq. (6)), hindering re-implementation.

4. Presentation hard errors: leftover template instructions in the main text; incorrect cross-reference in Fig. 4 (“Sec 4.2” vs §3.2); inconsistent/incorrect phrasing in Fig. 1 caption (“significant marginal benefit”).

5. Theory is framed too strongly for what is shown: Appendix F.1 is largely tautological; expert-diversity “guarantees” rely on toy assumptions (App. F.3) and should be softened or strengthened.

6. Reference hygiene issues: year/arXiv-ID mismatches, duplicates, and incomplete entries in the bibliography.

---

> ### Author Rebuttal · Authors · 2026-03-29
>
> We thank the reviewer and apologize for the citation/presentation errors. The BibTeX mismatch was due to metadata from the copied arXiv/Scholar BibTeX entry reflecting the latest revision rather than the original posting year, and the wrong figure/section pointer came from a late restructuring pass that deleted an earlier paragraph without updating a cross-reference. Both are our mistakes and will be fixed in revision.
>
> Abbrev.: T2/T3/F8=Tab.2/Tab.3/Fig.8; Q30/Q80=Qwen3-30B/Qwen3-Next-80B; A25/H25=AIME25/HMMT25.
>
> ## 1. Compute accounting
> Per-question means on 2xA6000-48GB. `Lat` is `x10^6 ms`. $N_{\text{inst}}=16$. Due to space, we show a subset.
>
> T2/T3/F8 use different anchors. In T2/F8, SC uses 1.5x initial width because it cannot adaptively expand, for fairer comparison to adaptive methods. T3 fixes width to isolate the expansion mechanisms with a separate SC baseline. F8 normalizes to fixed-width SC at $S_{\max}=80$, so the SC point is not 1.00. Compare within each block; across blocks, use the absolute columns, not `SC-norm`.
>
> |Src|Model|Data|Method|Acc|SC-norm|Abs tok|Eff tok|Lat|
> |---|---|---|---|---:|---:|---:|---:|---:|
> |T2|Q30|A25|SC|86.0|1.0|4.2|4.2|10.0|
> |T2|Q30|A25|DC|90.7|0.7|2.8|2.8|6.4|
> |T2|Q30|A25|RoE|84.7|1.6|3.0|6.9|7.9|
> |T2|Q30|A25|HyPER|95.3|0.7|2.1|3.0|6.0|
> |T3|Q30|A25|SC|86.7|1.0|3.2|3.2|7.7|
> |T3|Q30|A25|ST-only|93.3|1.8|2.8|5.8|7.2|
> |T3|Q30|A25|MT-only|90.0|1.9|6.1|6.1|14.0|
> |T3|Q30|A25|Manual|93.3|1.8|4.1|5.8|9.6|
> |T3|Q30|A25|HyPER|96.7|1.6|3.7|5.1|9.4|
> |F8|Q30|H25|SC@32|58.0|0.4|2.3|2.3|5.4|
> |F8|Q30|H25|DC@32|66.0|0.3|1.8|1.8|4.2|
> |F8|Q30|H25|RoE@32|65.0|0.7|1.9|3.9|5.2|
> |F8|Q30|H25|HyPER@32|68.0|0.35|1.5|2.0|4.7|
>
> `Abs tok`/`Eff tok`/`Lat` are in `x10^6`; `SC-norm` uses the SC anchor of each block. Under vLLM, `nvidia-smi` peak memory is similar across methods because `gpu_memory_utilization` drives pre-allocation; on 2xA6000-48GB, it is about 43 GB per GPU. We therefore report this real footprint; latency differences appear where HyPER/RoE may cost more per raw token but pruning still lowers end-to-end latency.
>
> ## 2. Table 6 sanity check
> We understand why this could look like copy/paste. But it is not: both means round to 23.6/30 = 78.7% over 5 runs. Qwen3-30B gives `23,24,24,23,24`; DeepSeek-8B gives `25,22,22,24,25`.
>
> ## 3. Warm-up normalization
> No dataset-level tuning is used. For each problem, we sample 16 warm-up self-consistency traces, compute the relevant statistic along decoding for each trace, record its maximum, and set $X_{\max}$ to the largest of those maxima for that problem; this value is then reused only for later online decisions on the same problem.
>
> App. A was unclear. "e.g., on a held-out dev set or the first few decisions" is stale text from an earlier draft and does not describe the pipeline. The experiments use per-problem warm-up normalization only, with no dev-set fitting, dataset-wide calibration, or manual tuning. We will correct this.
>
> ## 4. Window-level confidence in MULTITOKEN
> Let $c\_{p,i}$ denote the token-confidence quantity from Eq. (7), i.e., the competitor top-k negative average log-probability for token $i$ on path $p$. We define $C^{\mathrm{win}}\_{p}(t)=\frac{1}{L\_t}\sum\_{i=t-L\_t+1}^{t} c\_{p,i}$ and $L\_t=\min(W,t)$ with $W=2048$. In MULTITOKEN, after expanding a parent for $m$ steps, we evaluate each child by $C^{\mathrm{win}}\_{p}(t+m)$ and keep the highest.
>
> ## 5. Voting hyperparameter
> We use $K_a=3$.
>
> |Data|$K_a=2$|$K_a=3$|$K_a=4$|
> |---|---:|---:|---:|
> |A25|92.7|95.3|94.7|
> |H25|76.0|78.7|78.7|
>
> Failure-case analysis:
>
> |Data|Top-2 excludes but Top-3 keeps|Top-3 excludes but Top-4 keeps|
> |---|---:|---:|
> |A25|2/30|0/30|
> |H25|2/30|3/30|
>
> $K_a=4$ recovers some Top-3 misses but also adds noisy candidates, so there is no clear net gain.
>
> ## 6. "±", seeds, and randomness
> We clarify that the previous "±" did not denote std/std.err. It denoted the maximum absolute deviation from the mean across 5 independent stochastic runs:
> $\max\left(|\mathrm{acc}\_{\max}-\mathrm{acc}\_{\mathrm{avg}}|,\;|\mathrm{acc}\_{\min}-\mathrm{acc}\_{\mathrm{avg}}|\right)$
> In revision, we will replace it with standard deviation across the same 5 runs; conclusions are unchanged.
>
> All main results are averaged over 5 independent runs under fixed decoding settings (`temperature=0.6`, `top_p=0.95`, `top_k=40`), which `VLLMAdapter` passes to vLLM `SamplingParams`. No explicit seeds are fixed/logged yet. SC/path diversity comes from stochastic token sampling, and SINGLETOKEN adds Gumbel-noise-controlled refinement via `set_runtime_roe_hint(..., tau=0.3)`. We will document this more clearly in revision and expose, fix, and log explicit seeds.
>
> Finally, we agree Appendix F is overstated. As written, it suggests more justification than we currently provide. We will rethink this part carefully and aim to provide a more appropriate explanation and proof in the final version; for now it should be read as early intuition, not a complete formal justification.

---

> > ### Author Rebuttal · Reviewer_6mhe · 2026-04-02
> >
> > Thank you to the authors for the detailed rebuttal. The response was helpful and clarified several implementation details that were previously unclear. But several issues still require sharper clarification before I can consider them fully addressed.
> >
> > 1. **Compute accounting / fair budget comparison.**
> > The paper currently uses several related but non-identical notions of cost (e.g., normalized token count, effective tokens, instantiated paths), and the rebuttal provides only a representative subset of absolute numbers. In addition, the memory discussion mainly argues that peak memory looks similar under vLLM-style pre-allocation; this is informative in practice, but it does not fully answer the method-level scaling question about KV-cache / live-path overhead.
> >
> > 2. **Warm-up normalization and its implications.**
> > The clarification on Eq. (18) is helpful: the rebuttal states that \(X_{\max}\) is obtained per problem from 16 warm-up self-consistency traces, rather than from a held-out dev set. This addresses my earlier uncertainty about dataset-level tuning. However, this also means the method relies on a form of **per-instance online calibration**, which is conceptually different from having no calibration at all. It remains important to make clear whether this warm-up cost is fully included in the reported budget and whether the same protocol is applied consistently across methods.
> >
> > 3. **Definition of “window-level confidence” in MULTITOKEN.**
> > The rebuttal gives a much clearer definition of the window-level confidence used to rank child continuations, which I appreciate. But my remaining question is more about mechanism than notation: with the stated window size, it is still not fully clear how local/short-horizon this signal is in practice, especially on shorter or medium-length reasoning traces.
> >
> > 4. **Voting hyperparameter \(K_a\).**
> > The response clarifies that \(K_a=3\) is used and provides a small sensitivity analysis plus a failure-case count, which is helpful. However, I still do not have a complete picture of how robust this choice is beyond the two reported datasets, or whether the value was fixed a priori versus chosen after inspection of evaluation performance. Since answer truncation can directly affect the final claim about answer selection, this detail matters.*
> >
> > 5. **Randomness, variance reporting, and reproducibility.**
> > This remains the least resolved point for me. The rebuttal explains that the previous “±” notation was not standard deviation / standard error, and it also states that explicit seeds were not yet fixed/logged. I appreciate the authors’ transparency here, but proper uncertainty reporting is important, especially when some gains are moderate rather than overwhelming.*

---

> > > ### Author Response · Authors · 2026-04-04
> > >
> > > We thank the reviewer for the careful follow-up. We have prepared an anonymous repository with three tables: compute accounting, vote-truncation sensitivity, and fixed-seed reruns. Link: https://anonymous.4open.science/r/ICML_6mhe-ECAF/README.md
> > >
> > > **Compute accounting and fair budget comparison.**
> > > Table 1 in the link provides the raw data behind Tables 2, 3, and Figure 8; all values are per-question means. `SC-norm` is the relative metric used in the paper; `Abs Tokens` counts all generated tokens, including warm-up traces for DeepConf/HyPER; `Eff Tokens` additionally accounts for the MoE-equivalent routing multiplier in RoE/SingleToken computation. Thus mainly RoE/HyPER-like methods differ between Abs and Eff Tokens.
> > >
> > > We also add `Peak Active Paths`, `Avg Active Paths`, and an estimated `Peak KV Cache`. This KV estimate is algorithmic: for each trace step we compute `L_t=(live paths at t) x (current depth at t)`, then `L_peak=max_t L_t`; finally we convert it to GB by `Peak KV Cache ~= L_peak x (2 x n_layers x n_kv_heads x head_dim x 2 bytes) / 2^30`. For Qwen3-30B this is about `96 KiB` per live token. Multitoken steps can therefore have `Peak Active Paths > 80`, because an intermediate expansion with width `S` is counted as `S x ceil(80/S)` before the merge reduces it. This better captures method-level live-token residency than vLLM pre-allocation. The key relative pattern is clear: fixed-width baselines keep many long paths alive throughout decoding, while HyPER may have brief width spikes but still maintains a smaller sustained live set because continuous pruning reduces both active-path count and long-prefix residency. Even for SC, the KV bottleneck is determined by the number of simultaneously alive long paths, not by assuming every instantiated path reaches the longest final length and remains resident to the end.
> > >
> > > **Warm-up normalization and protocol consistency.**
> > > The 16 warm-up traces are an explicit per-instance online calibration stage. Their cost is fully included in our reported Absolute and Effective Tokens. These warm-up traces are executed before the main live pool and therefore contribute to token budget but not to concurrent KV residency. DeepConf uses the same warm-up protocol and we account for it the same way. Other baselines have no warm-up phase because they do not require online calibration or pruning.
> > >
> > > **Window-level confidence in MULTITOKEN.**
> > > Although the notation uses a global window, the ranking signal is local. Within one MULTITOKEN decision, all children expanded from the same root share the same prefix and the same normalization length. Therefore, the prefix token-conf contributes an identical constant offset and cancels under the argmax. For shorter traces, the effective window simply shrinks to the available context. Thus the child ranking is determined by the newly generated short-horizon continuation; we keep the same window-confidence form only to maintain a unified metric definition.
> > >
> > > **Voting hyperparameter $K_a$.**
> > > We fixed $K_a=3$ before the final benchmark sweep rather than selecting it post hoc per dataset. As shown in Table 2 of the linked repository, this choice remains stable across all 8 datasets: $K_a=2$ is consistently weaker, while keeping more low-frequency candidates beyond the top few usually brings little benefit and can introduce additional noise. This choice was informed by repeated observations from our DeepConf baseline runs and the corresponding ablations: when the correct answer appears within the top-3 vote candidates, it is usually still recoverable because the correct candidate often still exhibits relatively stable mean confidence and length statistics; once it falls outside the top few candidates, recovery becomes rare. We therefore used $K_a=3$ uniformly across all datasets as a simple fixed value that preserves most recoverable cases while trimming the noisy long tail.
> > >
> > > **Randomness, variance reporting, and reproducibility.**
> > > To address the reviewer’s concern directly, we re-ran the framework with 5 fixed global seeds (42, 43, 44, 45, 46) and report representative fixed-seed reruns in the linked Table 3 using the standard mean $\pm$ std convention. Importantly, these representative reruns were chosen to cover both large-margin settings and more moderate-gain settings, rather than only the most favorable cases. Across these reruns, the improvement direction remains consistent. On the harder reasoning benchmarks, the margin remains clearly larger than the run-to-run standard deviation; on smaller-margin settings, the gain is naturally narrower, which makes proper uncertainty reporting especially important and is precisely why we now provide fixed-seed mean $\pm$ std results. Due to rebuttal-time limits, we show representative seed-fixed reruns in the linked repository here; in the revision, we will update the main tables to follow the standard mean $\pm$ std convention and release the corresponding seed-fixed scripts for reproducibility.

---

### Official Review · Reviewer_tXgE · 2026-03-13

**Soundness:** 3
**Presentation:** 3
**Significance:** 2
**Originality:** 3
**Overall Recommendation:** 4
**Confidence:** 4

**Summary:**

The paper introduce HyPER a training-free, online control policy designed to optimize the trade-off between exploration  and exploitation during inference. Driven by the observation that the optimal balance between these two is phase-dependent, HyPER reallocates compute on the fly under a fixed token budget.

**Compliance With Llm Reviewing Policy:**

Affirmed.

**Key Questions For Authors:**

Please see the weakness.

**Limitations:**

Yes

**Strengths And Weaknesses:**

## Weakness

1.The design of the action selection although make sense but too hacky at the same time, following the paper idea there are inifinite many ways to do the selection, how can the paper make sure that this selection is the best. Is there any mathematical ground for the selection.

2. Following the previous point, there are too many hyparameters in the method which makes me worried about need heavy engineer effort to find a good set of hyperparameter. So it may limit the utility of this paper.

3. The contribution is limited.

## Strengths

1. The paper introduce a brand new standpoint by framing test-time scaling as a dynamic, closed-loop control problem, and moves tts beyond static schedules.

2. The paper provide really thourogh experiment and ablation study.

---

> ### Author Rebuttal · Authors · 2026-03-29
>
> We thank the reviewer for the careful reading and positive comments on our framing and results. We address three concerns: (1) whether the action design/selection is too heuristic, (2) whether HyPER has too many knobs, and (3) whether the contribution scope is limited.
>
> ## 1. On the action design and its grounding
> We agree that the reviewer is asking for a more principled justification of the controller. Rather than claiming a formal proof of global optimality, we demonstrate that HyPER provides a highly effective, principled, and interpretable solution to the online compute-allocation problem.
>
> The design principle is to match each action to the regime in which its marginal utility should be highest. The controller only needs to answer a local question: is the next unit of compute better spent on increasing coverage or refining promising paths? Our action scores are lightweight monotone surrogates for this decision. When diversity and consensus are both low, the pool is collapsing without agreement, so BRANCH has the highest marginal value. When confidence is low, entropy is high, and consensus is weak, the main issue is local instability rather than insufficient width, so SINGLETOKEN is favored because token-level refinement is the most direct exploitation primitive. When diversity is low but confidence dispersion across paths is high, MULTITOKEN becomes attractive because a short-horizon expand-reduce step can test nearby continuations and keep only the best child. Conversely, when the pool is already confident and sufficiently covered, the expected return of extra intervention is small, so NONE is preferred.
>
> In this sense, the signs in Eqs. (1)-(4) are not arbitrary, but a minimal training-free approximation to a harder online compute-allocation problem. We use simple equal-weight averaging because the goal is not to learn a complex policy, but to test whether coarse online signals suffice to coordinate exploration and exploitation. Thus, while we do not claim a proof of global optimality, the selector has a clear monotone rationale: each score increases when the corresponding action should become more useful.
>
> This interpretation is supported by our ablations. HyPER outperforms a manual early-explore/late-exploit schedule, suggesting that the gain comes from state-dependent online allocation rather than a fixed schedule. In revision, we will clarify this interpretation, soften stronger optimality wording, and explore whether a more formal justification can be added.
>
> ## 2. On hyperparameters
> We appreciate this concern, but in practice HyPER has fewer algorithmic degrees of freedom than it may appear.
>
> First, some quantities are budget/scale parameters, not HyPER-specific tuning knobs: ($S_{\max}, W$) and $N_{\text{init}}$ set the search scale.
>
> Second, some are structural scheduling parameters: the controller interval $T$ determines how often we re-evaluate the pool state.
>
> Third, the main algorithm-specific tunables are small in number: the diversity composition weight $\eta$ and the two-pass reuse penalty $\lambda$.
>
> Fourth, several quantities that may look tunable are actually automatic or fixed defaults, not manually engineered per dataset: the pruning threshold is determined automatically during warm-up; the $1/2$ and $1/3$ coefficients in the action scores are only normalization factors; and the expansion factor $r_t=\lceil W/|S_t|\rceil$ is derived online from the current survivor count.
>
> Importantly, App. E.1 shows that the main controller-related settings are robust: sweeping $T$ in {32, 64, 128} and $\eta$ in {0.2, 0.4, 0.8} causes only mild fluctuations, while sweeping $\lambda$ in {0.05, 0.1, 0.2} shows that 0.1 is a stable default and that both too-small and too-large values are slightly worse. This is the opposite of delicate benchmark-specific tuning.
>
> ## 3. On contribution scope
> We agree that the most durable contribution is not that every individual signal is new, but that TTS can be formulated as a dynamic online-control problem under a fixed budget, and that a simple training-free controller can coordinate multiple compute-allocation actions within this view.
>
> Concretely, the contribution is three-fold: (i) a framework-level reformulation of TTS as online expand-reduce control over a hypothesis pool; (ii) a practical training-free controller that unifies path-level exploration, short-horizon expand-reduce, and token-level refinement; and (iii) empirical evidence that this view improves the accuracy-compute trade-off over static schedules and pruning-only baselines across multiple backbones and benchmarks. We will revise the paper to make this scope more precise: HyPER provides a principled, highly effective, and interpretable training-free architecture that validates the immense potential of dynamic compute allocation for LLM reasoning, moving the field beyond rigid static schedules.

---

> > ### Author Rebuttal · Reviewer_tXgE · 2026-04-04
> >
> > Thank the authors for the thoughtful rebuttal. The response provides clarification on the design rationale of the controller, the practical robustness of the main hyperparameters. I will keep my current score.

---

> > > ### Author Response · Authors · 2026-04-08
> > >
> > > Thank you for your thoughtful follow-up and for recognizing the clarified design rationale and practical robustness of our method. We sincerely appreciate your time and consideration.

---

### Decision · Program_Chairs · 2026-04-30

**Decision:**

Accept (regular)

**Comment:**

This paper proposes HyPER, a training-free framework for test-time scaling that formulates inference as a dynamic online control problem and adaptively allocates computation between exploration and exploitation under a fixed budget. The reviewers generally found this framing novel and timely, and they viewed the empirical study and ablations as a meaningful strength of the paper. The main concerns centered not on the overall value of the idea, but on the degree of principled grounding and reporting rigor: in particular, reviewers questioned whether the controller is too heuristic, whether some design choices and hyperparameters are insufficiently justified, and whether the paper’s compute accounting, reproducibility details, and theoretical claims are stated as carefully as they should be. The rebuttal addressed a substantial portion of these issues by clarifying the controller rationale, providing additional accounting details, and softening some claims, although some concerns about rigor and presentation remain. On balance, I find the paper to make a useful and empirically supported contribution, and I lean accept.